# SALL3 expression balance underlies lineage biases in human induced pluripotent stem cell differentiation

Takuya Kuroda[1], Satoshi Yasuda [1], Shiori Tachi[1,2], Satoko Matsuyama[1,3], Shinji Kusakawa[1], Keiko Tano[1], Takumi Miura[1], Akifumi Matsuyama[3] & Yoji Sato[1,2,4,5]

Clinical applications of human induced pluripotent stem cells (hiPSCs) are expected, but hiPSC lines vary in their differentiation propensity. For efficient selection of hiPSC lines suitable for differentiation into desired cell lineages, here we identify *SALL3* as a marker to predict differentiation propensity. *SALL3* expression in hiPSCs correlates positively with ectoderm differentiation capacity and negatively with mesoderm/endoderm differentiation capacity. Without affecting self-renewal of hiPSCs, *SALL3* knockdown inhibits ectoderm differentiation and conversely enhances mesodermal/endodermal differentiation. Similarly, loss- and gain-of-function studies reveal that SALL3 inversely regulates the differentiation of hiPSCs into cardiomyocytes and neural cells. Mechanistically, SALL3 modulates DNMT3B function and DNA methyltransferase activity, and influences gene body methylation of Wnt signaling-related genes in hiPSCs. These findings suggest that SALL3 switches the differentiation propensity of hiPSCs toward distinct cell lineages by changing the epigenetic profile and serves as a marker for evaluating the hiPSC differentiation propensity.

[1] Division of Cell-Based Therapeutic Products, National Institute of Health Sciences, 3-25-26 Tonomachi, Kawasaki-ku, Kawasaki 210-9501, Japan. [2] Department of Quality Assurance Science for Pharmaceuticals, Graduate School of Pharmaceutical Sciences, Nagoya City University, 3-1 Tanabe-dori, Mizuho-ku, Nagoya 467-8603, Japan. [3] Department of Regenerative Medicine, School of Medicine, Fujita Health University, 1-98 Dengakugakubo Kutsukake-cho, Toyoake, Aichi 470-1192, Japan. [4] Department of Cellular & Gene Therapy Products, Graduate School of Pharmaceutical Sciences, Osaka University, 1-6 Yamadaoka, Suita, Osaka 565-0871, Japan. [5] Department of Translational Pharmaceutical Sciences, Graduate School of Pharmaceutical Sciences, Kyushu University, 3-1-1 Maidashi, Higashi-ku, Fukuoka 812-8582, Japan. Correspondence and requests for materials should be addressed to Y.S. (email: yoji@nihs.go.jp)

Human pluripotent stem cells (hPSCs) have the ability to differentiate into various types of cells and to self-renew in vitro. Because of these two characteristics, hPSCs are expected to provide new applications for regenerative medicine/cell therapy. In recent years, many clinical-grade human embryonic stem cell (hESC) lines and human-induced pluripotent stem cell (hiPSC) lines have been established for use as raw materials for cell-based therapeutic products (CTPs)[1,2]. Essentially, hESC and hiPSC lines exhibit variation in their individual differentiation propensities for generating specific cell lineages during in vitro differentiation[3–5]. Therefore, selection of hPSC lines capable of efficiently differentiating into target cells is quite important for the practical application of CTPs.

To understand line-to-line variation, including differences in differentiation propensity, previous studies have performed large-scale validations of hPSC lines and revealed distinct epigenetic and transcriptional profiles among hPSC lines[6,7]. However, until now, only a few markers to predict the differentiation propensity of hPSC lines have been reported. miR-371-3 has both a predictive and functional role in hPSC neurogenic differentiation behavior, and its expression at the undifferentiated stage predicts the neural differentiation propensity of hPSCs[8]. Hematopoietic commitment of hPSCs to hematopoietic precursors correlates with the IGF2 expression level, which depends on signaling-dependent chromatin accessibility[9]. Non-CG DNA methylation at several sites is proposed as a biomarker for assessing endodermal differentiation capacity in hPSCs[10]. To our knowledge, no marker has been identified to simultaneously predict the differentiation propensity of hPSCs toward multiple cell lineages.

In this study, we identify the *SALL3* gene as a marker predictive of differentiation propensity, using the rank correlation method and analysis of ten hiPSC lines. The *SALL3* expression correlates positively with ectoderm differentiation and negatively with mesoderm/endoderm differentiation during embryoid body (EB) formation. In addition, SALL3 inversely regulates the capacities of cardiac and neural differentiation in hiPSCs. Mechanistically, SALL3 is found to repress gene body methylation in hiPSCs, leading to their epigenetic changes. These findings provide a practical method for selecting appropriate hPSC lines in clinical-grade cell banks, allowing the prediction of differentiation capacity toward a desired cell lineage.

## Results

**Profiles of hiPSC lines showing differentiation propensities.** Hypothesizing that some critical attribute in hiPSCs underlies the determination of propensity to differentiate into a specific lineage, we attempted to find potential marker genes, the expression of which in hiPSCs significantly correlated with the efficacy of differentiation into three germ layers. Our approach for identifying differentiation propensity markers is essentially based on the statistical comparison of the gene-expression profiles of undifferentiated hiPSCs with each cell line's in vitro differentiation potential using the rank correlation method (Fig. 1a). First, ten hiPSC lines were cultured for several passages under feeder-free conditions, and we examined their comprehensive transcriptional profiles using microarray analysis. The defined filtering criteria (see Methods) identified a set of 3362 probes with significantly different expression levels among ten hiPSC lines (Fig. 1b, Supplementary Data 1).

Next, to profile differences in the ability to differentiate into the three germ layers, we examined differentiation propensity of ten hiPSC lines via spontaneous differentiation during EB formation without addition of growth factors to minimize any bias toward differentiation. Using total RNA isolated from the EBs after 16 days of differentiation, we obtained the transcript expression profiles of total 97 genes comprised of 45 ectoderm markers, 56 mesoderm markers, and 27 endoderm markers, which included common markers for two or three germ layers (Supplementary Figure 1 and Supplementary Data 2). To reduce the number of variables indicating each cell line's differentiation propensity, gene-expression data were analyzed with principal component analysis (PCA), and we calculated the first principal component score (PC1) for the three germ layers. Based on the results of PC1, we ranked the ten hiPSC lines in descending order of PC1 for each germ layer (Fig. 1c). Interestingly, we observed a significant negative correlation between the ectoderm and mesoderm differentiation ranks ($r = -0.66$, $P < 0.05$) and a significant positive correlation between the mesoderm and endoderm differentiation ranks ($r = 0.79$, $P < 0.02$). Thus, under the nondirected EB differentiation of hiPSCs, a strong propensity to differentiate into mesoderm indicated a weak propensity to differentiate into ectoderm but a strong propensity to differentiate into endoderm, and vice versa (Fig. 1d).

Finally, Spearman's rank correlation coefficients were determined between the gene-expression rank (microarray data) and the ectoderm/mesoderm/endoderm differentiation rank (PC1). As a result, we identified differentiation propensity marker candidates that indicated statistically significant correlations ($P < 0.05$). The Spearman correlation coefficient can describe both positive and negative correlations. There were 90-positive and 70-negative genes, 20-positive and 7-negative genes, and 7-positive and 25-negative genes for ectoderm, mesoderm, and endoderm differentiation markers, respectively (Fig. 2a and Supplementary Data 3). The top five differentiation propensity marker candidate genes are summarized in Table 1.

**SALL3 is a marker for differentiation propensity of hiPSCs.** We next investigated which genes among our identified marker candidates were relevant for predicting differentiation propensity. For this purpose, we identified genes functionally important for differentiation propensity among the candidates. On the basis of our observation that the differentiation efficiency of hiPSC lines toward ectoderm conflicted with that toward the mesoderm/endoderm (Fig. 1d), we hypothesized that proper functional markers for differentiation propensity should exhibit an inverse correlation between ectoderm and mesoderm/endoderm differentiation. To test this hypothesis, we surveyed genes showing a positive (negative) correlation in ectoderm differentiation and a negative (positive) correlation in mesoderm/endoderm differentiation. Among the identified marker candidates, *SALL3* was the only gene satisfying this inverse correlation. Thus, *SALL3* expression positively correlated with ectoderm differentiation propensity and negatively correlated with mesoderm/endoderm differentiation propensity (Fig. 2a). The expression of *SALL3* mRNA in ten hiPSC lines was almost linearly dispersed among an approximately fivefold range (Fig. 2b). To further confirm our hypothesis that *SALL3* could be a marker for differentiation propensity, we additionally employed other two hiPSC lines, 606A1 and 648A1, as a test set and differentiated these hiPSCs into EBs. In an undifferentiated state, mRNA levels of *SALL3* in 606A1 hiPSCs were significantly higher than those in 648A1 hiPSCs (Supplementary Figure 2A). As expected from the *SALL3*-expression levels in hiPSC lines, EBs derived from 606A1 hiPSCs exhibited higher expression of ectoderm marker genes (*PAX6*, *NES*, and *SOX1*), and lower expression of both mesoderm marker genes (*GATA4*, *T*, and *KDR*) and endoderm marker genes (*SOX7*, *SOX17*, and *GATA6*), compared with 648A1 hiPSCs (Supplementary Figure 2B–D).

To verify the roles of SALL3 in regulating the differentiation propensities of hiPSCs, we performed a loss-of-function

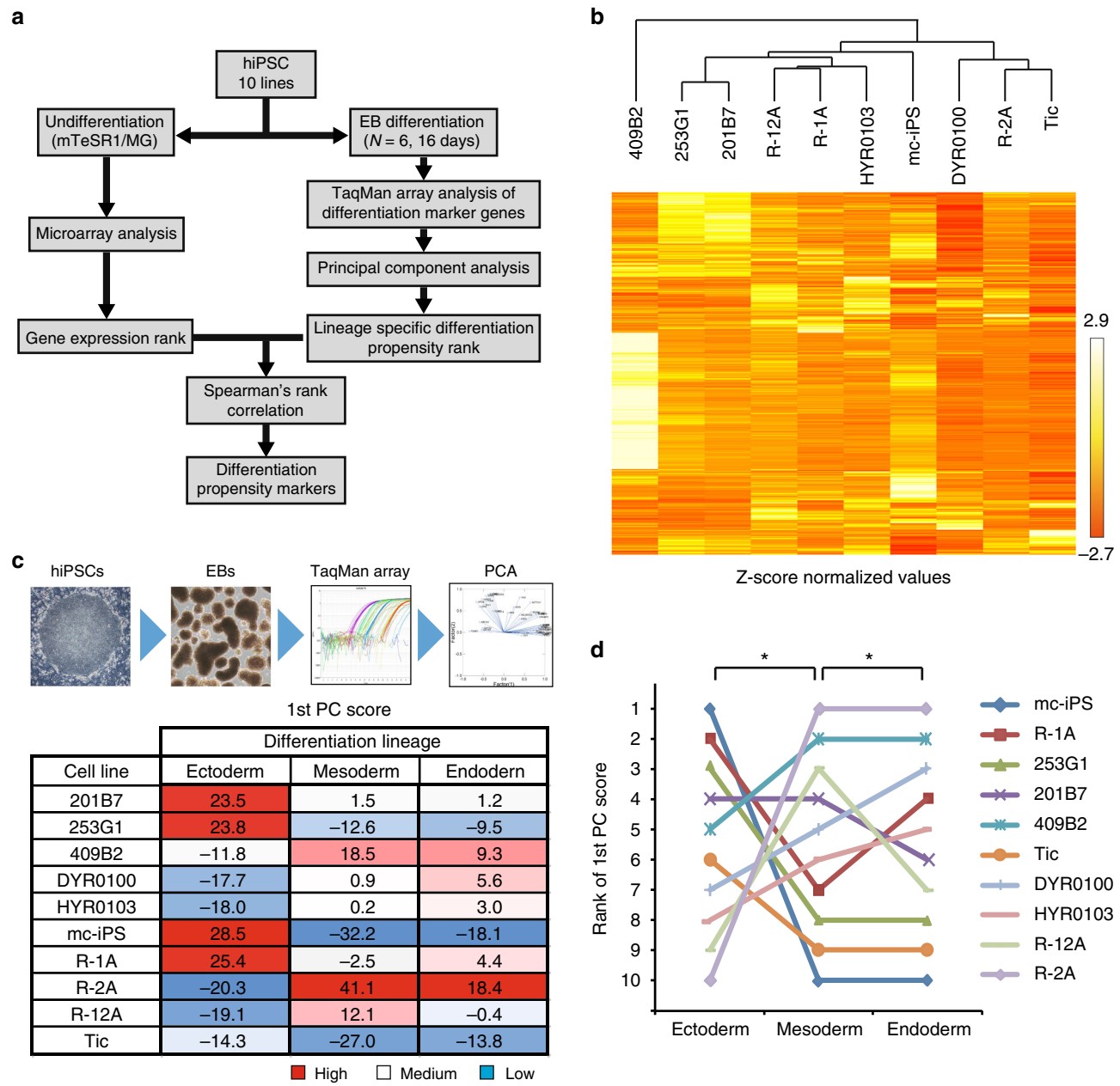

**Fig. 1** Profiles of hiPSC lines showing differentiation propensities. **a** Outline of workflow for identification of biomarkers capable of predicting the differentiation propensity of hiPSCs. **b** Hierarchical clustering of gene expression in ten hiPSC lines. We identified 3362 probes with significantly different expression levels among ten hiPSC lines. **c** Expression profiles for lineage marker genes were summarized using PCA. The number indicates PC1 of each lineage among the ten hiPSC lines. **d** The line graph represents the rank of the first PC score of each lineage among the ten hiPSC lines. *$P < 0.05$, Spearman's rank correlation coefficients

experiment by transducing 253G1 cells with lentiviral vectors containing shRNA for *SALL3*. The knockdown (KD) efficiency of the generated *SALL3* KD cells was confirmed by expression of SALL3 transcript and protein (Fig. 2c, d). We also observed that *SALL3* KD poorly affected the mRNA expression of pluripotency markers (*OCT3/4* and *NANOG*) in undifferentiated hiPSCs (Fig. 2e). We then subjected *SALL3* KD cells to EB formation and examined their differentiation into three germ layers. EBs derived from *SALL3* KD cells exhibited significantly lower expression of ectoderm marker genes (*PAX6*, *NES*, and *SOX1*) and, conversely, significantly higher expression of both mesoderm marker genes (*GATA4*, *T*, and *KDR*) and endoderm marker genes (*FOXA2* and *AFP*) compared with levels in control cells (Fig. 2f–h). Notably, we obtained similar results following EB

formation from hiPSCs transduced with another clone of *SALL3* shRNA, excluding off-target effects of *SALL3* KD on differentiation (Supplementary Figure 3). These data indicate that *SALL3* expression in hiPSCs positively regulates ectoderm differentiation and negatively regulates mesoderm/endoderm differentiation, which is consistent with the Spearman's rank correlation coefficients of *SALL3* (Fig. 2a and Supplementary Data 3).

**SALL3 inversely regulates cardiac and neural differentiation.** Our results demonstrated that suppression of SALL3 was sufficient to switch the differentiation propensity of hiPSCs during nondirected EB differentiation. We next examined the functional role of SALL3 in directed differentiation into cardiomyocytes as

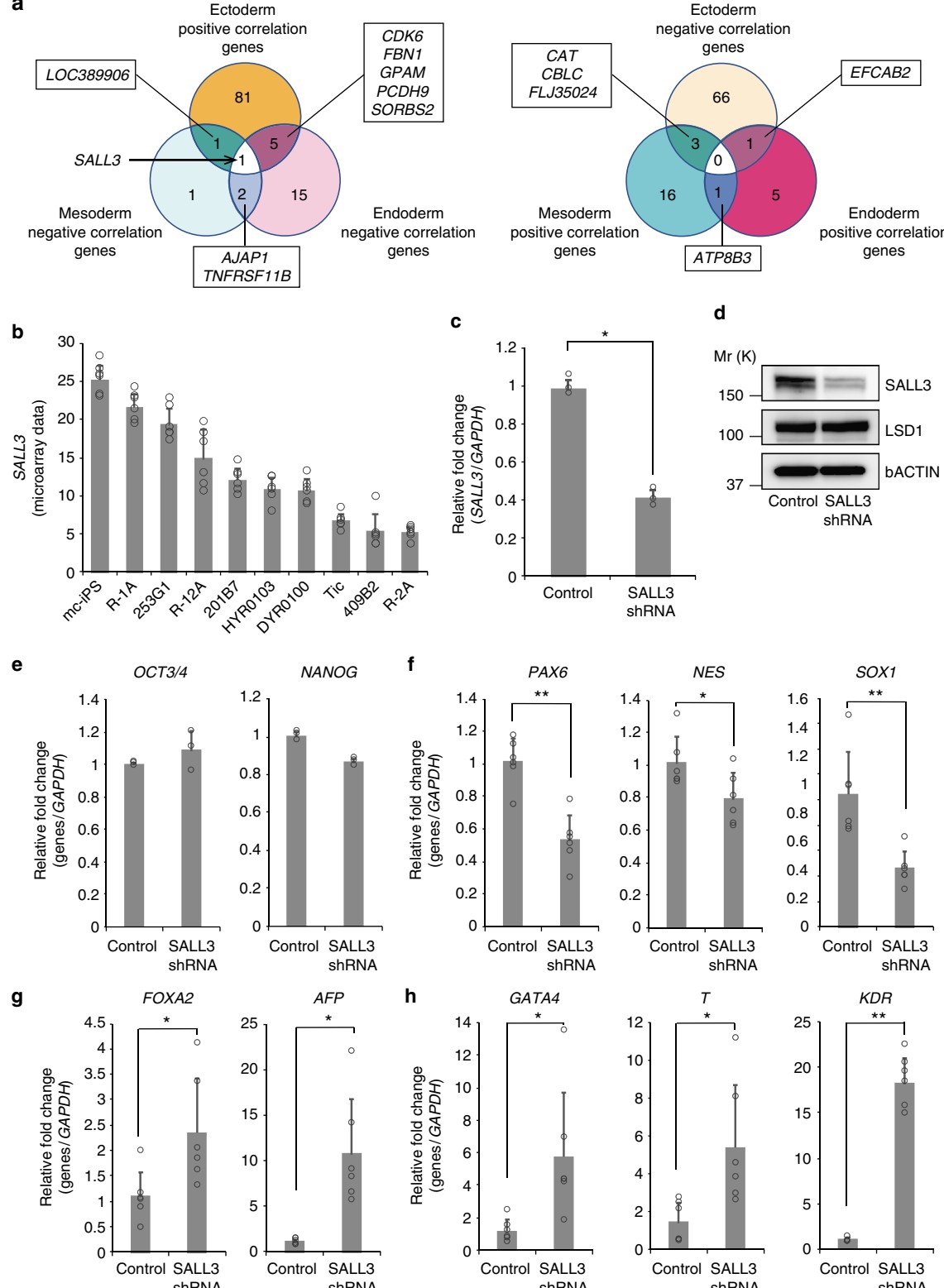

an example of a mesoderm lineage. hiPSCs were differentiated into cardiomyocytes using an in vitro differentiation protocol, as previously described[11]. hiPSCs were temporally treated with a GSK3 inhibitor, CHIR99021, on days 0–1 to activate Wnt signaling, followed by the addition of a Wnt inhibitor, IWP4, on days 3–5 (Fig. 3a). Cardiomyocyte differentiation in hiPSCs was evaluated by the expression of representative markers for cardiac progenitors and cardiomyocytes using quantitative real-time

polymerase chain reaction (qRT-PCR) and flow cytometry. As expected, the *SALL3* KD cells exhibited altered marker profiles, with dramatic increases in the expression of *GATA4*, *NKX2.5*, and *TNNT2* (Fig. 3b). Flow cytometry analysis showed that the population of TNNT2-positive cells increased to 63.2% in *SALL3* KD cells compared with 19.5% in control shRNA-treated cells (Fig. 3c). These results show that the suppression of SALL3 markedly enhances cardiomyocyte differentiation in hiPSCs,

**Fig. 2** SALL3 regulates differentiation into the three germ layers. **a** Venn diagrams illustrate overlaps among differentiation propensity marker candidate genes. *SALL3* was the only gene showing an inverse correlation between ectoderm and mesoderm/endoderm differentiation. All candidate genes of differentiation propensity markers were listed in Supplementary Data 3. **b** Microarray data of *SALL3* expression in ten hiPSC lines ($n = 6$, biological replicates). **c** *SALL3* knockdown was confirmed by qRT-PCR analysis ($n = 3$, biological replicates). $^*P < 0.01$, two-sided $t$ test. **d** Western blot analysis of the total extracts obtained from control and *SALL3* knockdown cells. LSD1 and β-actin were used as a nuclear protein control and loading control, respectively. Molecular weight is indicated as Mr (k). **e** qRT-PCR analysis of undifferentiated hPSC markers, *OCT3/4* and *NANOG*. Total RNA was isolated from 253G1 *SALL3* shRNA cells and 253G1 control shRNA cells in the undifferentiated state ($n = 3$, biological replicates). **f–h** qRT-PCR analysis of three germ layer-specific genes in EBs derived from 253G1 *SALL3* shRNA cells and 253G1 control shRNA cells ($n = 6$, biological replicates). Ectoderm marker genes (**f**), endoderm marker genes (**g**), and mesoderm marker genes (**h**) are shown. $^*P < 0.05$, $^{**}P < 0.01$, two-sided $t$ test. Error bars represent mean ± SD

### Table 1 Top five propensity marker candidate genes

| Ectoderm | | | | Mesoderm | | | | Endoderm | | | |
|---|---|---|---|---|---|---|---|---|---|---|---|
| Positive correlation | | Negative correlation | | Positive correlation | | Negative correlation | | Positive correlation | | Negative correlation | |
| Gene | $r_s$ | Gene | $r_s$ | Gene | $r_s$ | Gene | $r_s$ | Gene | $r_s$ | Gene | $r_s$ |
| LOC389906 | 0.90 | UBR5 | −0.87 | LOC400680 | 0.87 | TNFRSF11B | −0.79 | DDX58 | 0.84 | TNFRSF11B | −0.89 |
| ADI1 | 0.85 | CPZ/GPR78 | −0.85 | TRIM4 | 0.81 | EN2 | −0.78 | C3orf67 | 0.73 | NXPH2 | −0.81 |
| NUP98 | 0.84 | LY75 | −0.83 | ALPK3 | 0.79 | AJAP1 | −0.72 | CRYZ | 0.72 | RAB3B | −0.79 |
| LUC7L3 | 0.83 | C6orf54 | −0.83 | KLK5 | 0.79 | FOXG1 | −0.70 | AACSP1 | 0.68 | ST8SIA4 | −0.79 |
| WNT10B | 0.82 | C3 | −0.83 | ALPK3 | 0.79 | ASCL1 | −0.68 | SLC25A4 | 0.68 | NHS | −0.78 |

which is consistent with the results of experiments using another hiPSC line (Supplementary Figure 4). To further confirm the role of SALL3 in cardiomyocyte differentiation, we transduced 253G1 cells with lentiviral vector for EF1α promoter-driven expression of the *SALL3* gene to establish *SALL3* overexpressing cells. The *SALL3* mRNA levels in the overexpression cells were approximately 2.3-fold higher than those in control cells (Fig. 3d). In contrast, *SALL3* overexpressing cells exhibited marked decreases in the transcript levels of *GATA4*, *NKX2.5*, and *TNNT2* compared to levels in control cells after the induction of cardiomyocyte differentiation in hiPSCs (Fig. 3e). These results from gain-of-function experiments also suggest that SALL3 is a negative regulator of hiPSCs for cardiomyocyte differentiation without a step of EB formation.

The SALL3 regulation of cardiomyocyte differentiation in hiPSCs allowed us to examine whether SALL3 expression influenced the direct differentiation into neural cells, an ectoderm lineage. Neural differentiation was performed by dual inhibition of SMAD signaling as described previously[12]. Namely, hiPSCs were treated with Noggin on days 0–10 and with SB531542 on days 0–6 under a gradual increase in N2 medium concentration (Fig. 4a). *SALL3* KD cells after the neural differentiation showed a significant decrease in the mRNA expression of markers for neural ectoderm (*PAX6* and *SOX1*), neural progenitors (*NES*), and dopaminergic neurons (*TH*) compared with levels in control cells (Fig. 4b). Similar results were obtained with *SALL3* KD cells derived from another hiPSC line (Supplementary Figure 5). Inversely, *SALL3* overexpressing cells clearly exhibited enhanced mRNA expression of neural cell markers after differentiation (Fig. 4c). Immunofluorescence images also indicated down and upregulation of PAX6 protein expression in *SALL3* KD cells and *SALL3* overexpressing cells, respectively, after the induction of neural differentiation in hiPSCs. In contrast to PAX6 expression, expression of the pluripotency marker OCT3/4 apparently disappeared in *SALL3* overexpressing cells after the neural differentiation (Fig. 4d). Furthermore, we also confirmed improvement of neural differentiation by overexpressing *SALL3* in a strain R-2A that had shown the lowest *SALL3* expression and

propensity for ectoderm differentiation among the ten hiPSC lines (Supplementary Figure 6). These results indicate that SALL3 positively regulates the direct differentiation of hiPSCs into neural cells. Our findings strongly support the idea that SALL3 is particularly crucial as a switch of differentiation propensity in hiPSCs.

**SALL3 modulates hiPSC differentiation via DNMT3B.** Epigenetic features in cells, such as DNA methylation profiles, are broadly known to play an important role in determining cell characteristics[13]. SALL3 was previously reported to interact with DNMT3A and to inhibit CpG island methylation in hepatocellular carcinoma[14]. Therefore, we examined the endogenous interaction of SALL3 with the de novo DNA methyltransferases DNMT3A and DNMT3B, and a maintenance enzyme DNMT1 in hiPSCs. Immunoprecipitation analysis with cell lysates using anti-SALL3 antibody showed that DNMT3B clearly co-precipitated with SALL3, while neither DNMT3A nor DNMT1 co-precipitated with SALL3 (Fig. 5a). Immunoprecipitation using anti-DNMT3B antibody also confirmed the physical interaction between SALL3 and DNMT3B (Fig. 5b). In order to examine the effect of SALL3 on DNA methyltransferase (DNMT) activity in hiPSCs, we knocked out *SALL3* gene in 253G1 cells using a genome-editing technique and measured DNMT activity of the nuclear fraction prepared from the clone in vitro. The knockout of *SALL3* gene markedly increased endogenous DNMT activity of hiPSCs, suggesting the inhibitory role of SALL3 in molecular function of DNMTs (Fig. 5c), though the quantitative measurement of the endogenous DNMT3B isoform activity after immunoprecipitation with anti-DNMT3B antibody was not successful, presumably due to the insufficient amount or stability of DNMT3B protein for the assay we employed. Thus, we further investigated whether DNMT3B functionally contributed to the changes in differentiation propensity by the *SALL3* deficiency in hiPSCs. By applying a CRISPR gRNA that targets the sequence in the second exon containing the start codon of *DNMT3B* in *SALL3*$^{-/-}$ cell, we tried to generate *SALL3/DNMT3B* double-

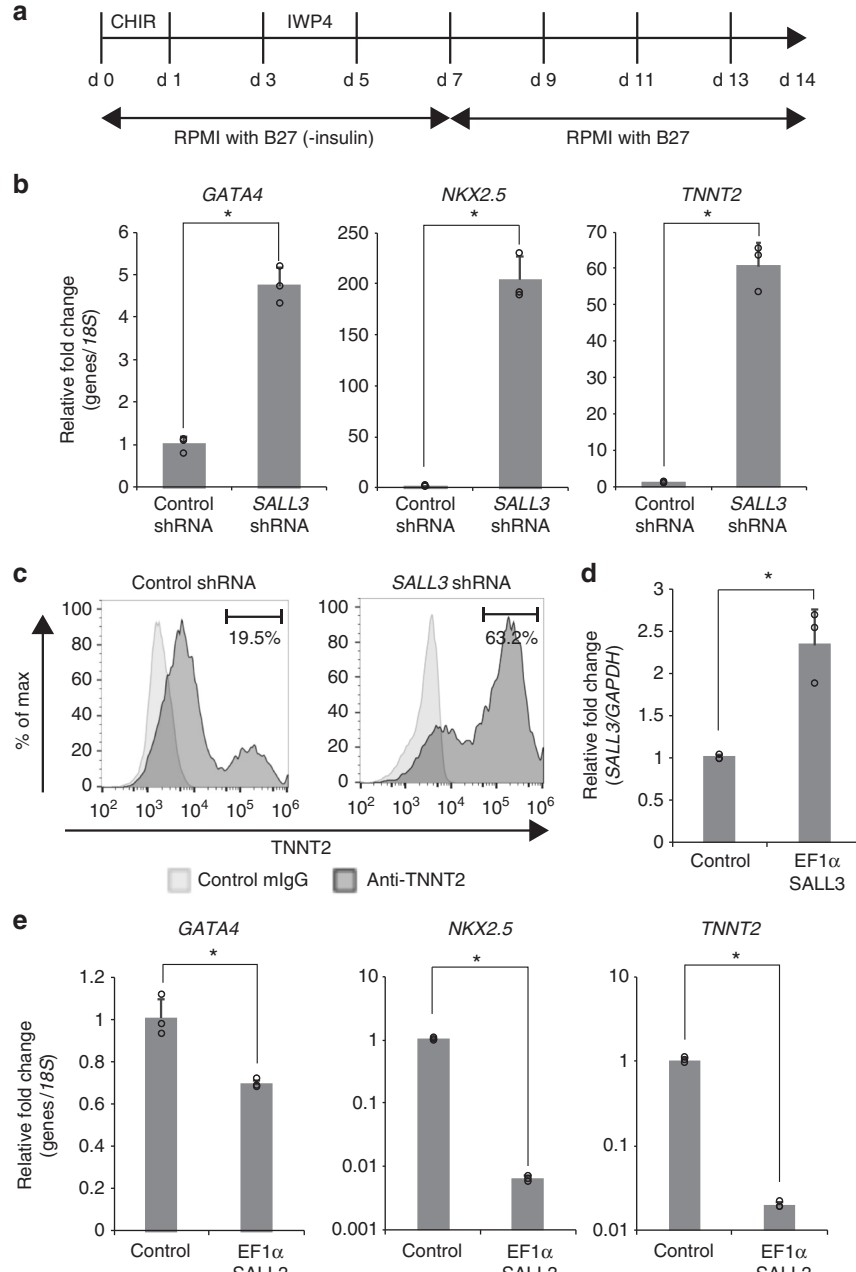

**Fig. 3** *SALL3* knockdown accelerates cardiac differentiation. **a** Schematic diagram of culture procedures for cardiomyocyte. **b** qRT-PCR analysis of cardiomyocyte markers *GATA4*, *NKX2.5*, and *TNNT2*. Total RNA was isolated from 253G1 *SALL3* shRNA cell derived and 253G1 control shRNA cell-derived cardiomyocytes (*n* = 3, biological replicates). *$P$ < 0.01, two-sided *t* test. **c** Flow cytometry analysis of TNNT2 in 253G1 *SALL3* shRNA cell-derived (right) and 253G1 control shRNA cell-derived (left) cardiomyocytes. **d** *SALL3* mRNA levels in undifferentiated 253G1 *SALL3* overexpressing cells (EF1α-SALL3). *n* = 3, biological replicates. *$P$ < 0.01, two-sided *t* test. **e** qRT-PCR analysis of cardiomyocyte markers *GATA4*, *NKX2.5*, and *TNNT2*. Total RNA was isolated from 253G1 EF1α-*SALL3* cell-derived and 253G1 control vector cell-derived cardiomyocytes (*n* = 3, biological replicates). *$P$ < 0.01, two-sided *t* test. Error bars represent mean ± SD

knockout lines. However, despite identifying the biallelic mutations in DNMT3B gene, we were unable to obtain any homozygous DNMT3B knockout hiPSC clones. It is presumably due to growth inhibition and/or lethality caused by complete depletion of DNMT3B protein as DNMT3B knockout hESC lines were successfully established with exon 21 DNMT3B mutations[15]. Therefore, among the clones obtained, we employed *SALL3*$^{-/-}$ *DNMT3B*$^{-/mut}$ clone, which expresses protein encoded by a heterozygous *DNMT3B* gene with a single nucleotide substitution and an eighteen base addition mutation in the *SALL3* knockout

background (Supplementary Figure 7A), for further experiments. In *SALL3*$^{-/-}$ cells, the expression of DNMT3B protein was comparable to the wild type's, whereas DNMT3B was downregulated approximately by 60% in *SALL3*$^{-/-}$ *DNMT3B*$^{-/mut}$ cells (Supplementary Figure 7B). We next performed directed differentiation into neural cells and cardiomyocytes using *SALL3*$^{-/-}$ *DNMT3B*$^{-/mut}$ cells. The introduction of *DNMT3B*$^{-/mut}$ showed partial rescue of the attenuated neural differentiation observed in SALL3$^{-/-}$ cells, when we examined the mRNA expression of *PAX6* and *SOX1* (Fig. 5d). Furthermore,

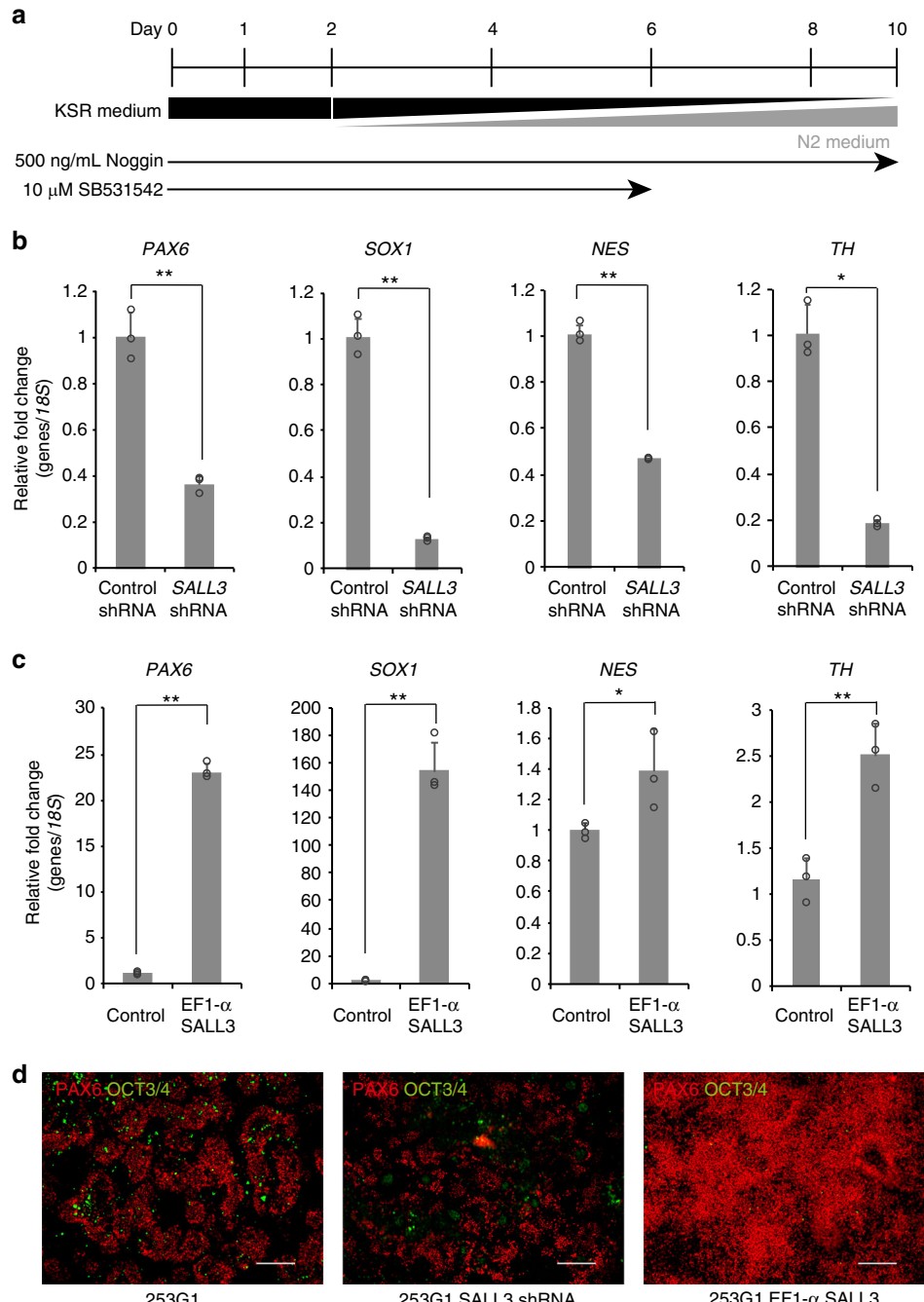

**Fig. 4** *SALL3* expression regulates neural differentiation. **a** Schematic diagram of culture procedures for neural cells. **b, c** qRT-PCR analysis of neural cell markers *PAX6*, *SOX1*, *NES*, and *TH*. Total RNA from 253G1 *SALL3* shRNA cell-derived and 253G1 control shRNA cell-derived neural cells (**b**) and 253G1 EF1α-*SALL3* cell-derived and 253G1 control vector cell-derived neural cells (**c**) was isolated ($n = 3$, biological replicates). $^*P < 0.05$, $^{**}P < 0.01$, two-sided $t$ test. **d** Immunofluorescence staining of PAX6 and OCT3/4 in 253G1 cell-derived (left), 253G1 *SALL3* shRNA cell-derived (center), and 253G1 EF1α-*SALL3* cell-derived neural cells (right). Scale bars, 200 μm. Error bars represent mean ± SD

$SALL3^{-/-}$ $DNMT3B^{-/\mathrm{mut}}$ cells significantly suppressed the enhanced cardiomyocyte differentiation of $SALL3^{-/-}$ cells accompanied with the change of the expression of *GATA4*, *NKX2.5*, and *TNNT2* (Fig. 5e). Our results suggest that DNMT3B is functionally contributes to the lineage differentiation phenotypes observed in *SALL3*-depleted hiPSCs.

**SALL3 changes gene-body methylation patterns in hiPSCs.**
Next, we determined whether SALL3 affected DNA methylation patterning at hiPSC CpG sites. Using the Illumina Infinium

HumanMethylation450 (HM450) DNA methylation array, we comprehensively compared the DNA methylation landscape of SALL3 KD 253G1 cells with that of control shRNA-transduced 253G1 cells. A significant change in the methylation status was observed in 3.66 % of the total probes (Fig. 6a, left), with the majority (85.65 %) of differentially methylated probes being hypermethylated in the *SALL3* KD sample (Fig. 6a, right), which suggests that *SALL3* KD promotes methylation of CpG methylation sites in the genome of hiPSCs. Collectively, our results suggest that SALL3 inhibits CpG methylation via DNMT3B in

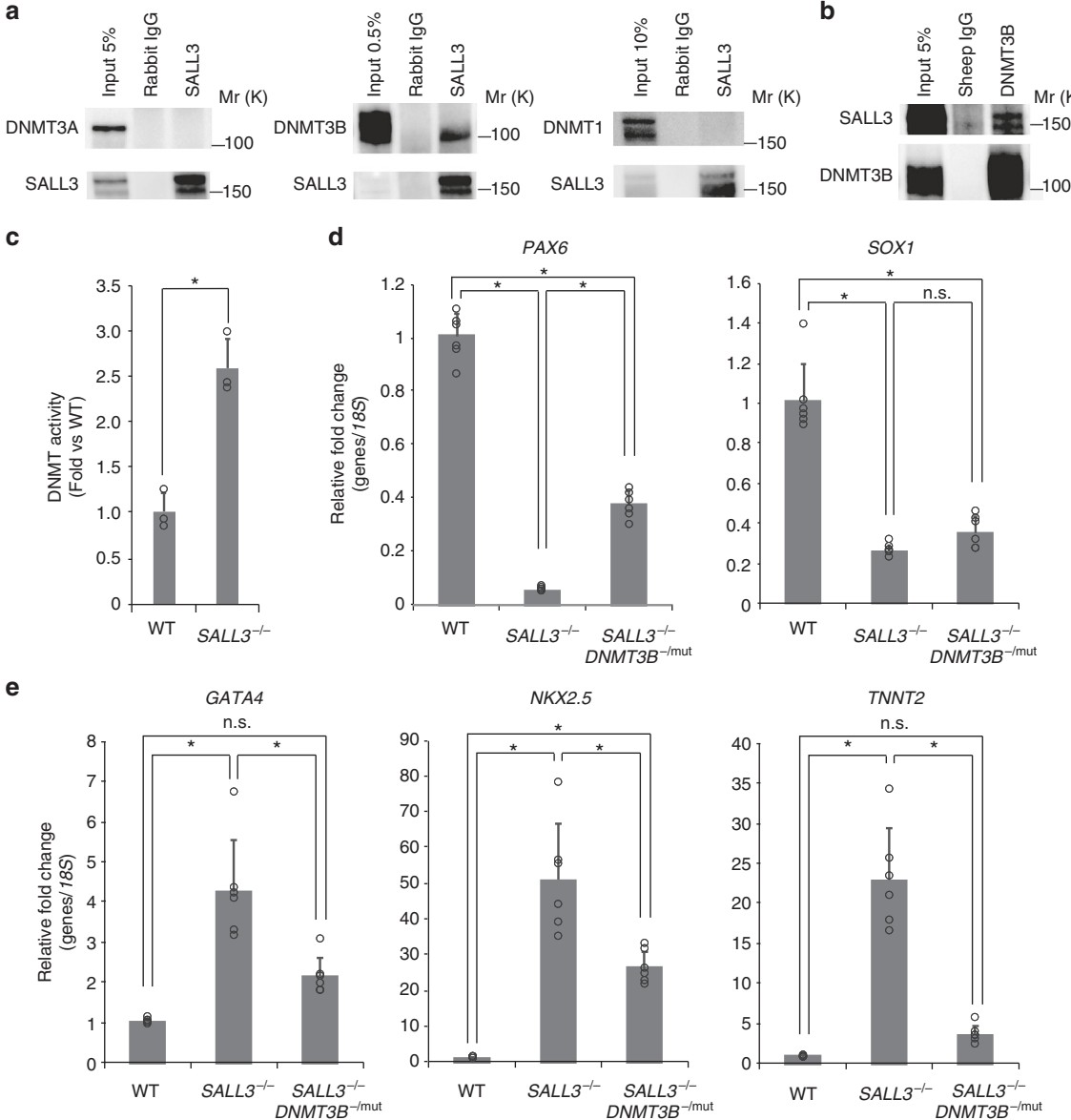

**Fig. 5** SALL3 interacts with DNMT3B and modulates DNMT function. **a** Lysate prepared from 253G1 cells was immunoprecipitated with anti-SALL3 or normal rabbit IgG. The immunoblot was analyzed with anti-DNMT3A, DNMT3B, and DNMT1 antibodies. Molecular weight is indicated as Mr (k). **b** Lysate prepared from 253G1 cells was immunoprecipitated with anti-DNMT3B or normal sheep IgG. The immunoblot was analyzed with anti-SALL3 antibody. **c** In vitro DNMT activity assay. DNMT activity of the nuclear fractions prepared from 253G1 (WT) cells and the 253G1 SALL3$^{-/-}$ cells was measured (n = 3, biological replicates). *P < 0.01, two-sided t test. **d** qRT-PCR analysis of neural cell markers PAX6 and SOX1. Total RNA was isolated from 253G1-derived (WT), 253G1 SALL3$^{-/-}$-derived and 253G1 SALL3$^{-/-}$ DNMT3B$^{-/mut}$-derived neural cells (n = 6, biological replicates). *P < 0.01, one-way ANOVA with post hoc Tukey–Kramer test. **e** qRT-PCR analysis of cardiomyocyte markers GATA4, NKX2.5, and TNNT2. Total RNA was isolated from 253G1-derived (WT), 253G1 SALL3$^{-/-}$-derived and 253G1 SALL3$^{-/-}$ DNMT3B$^{-/mut}$-derived cardiomyocytes (n = 6, biological replicates). *P < 0.01, one-way ANOVA with post hoc Tukey–Kramer test. Error bars represent mean ± SD

hiPSCs. Recently, several studies have demonstrated that DNMT3B is responsible for DNA methylation within gene bodies[16]. Therefore, we next divided the hypermethylated probes into seven categories according to their genomic positions and found that gene body regions (27.09 %) comprised the category most enriched in CpG methylation (Fig. 6b). To analyze the functional enrichment of the genes containing hypermethylated probes in the gene body region (Supplementary Data 4), we performed bioinformatics pathway analysis using IPA software. IPA revealed that these genes were associated with the pathways of axonal guidance signaling, neuropathic pain signaling in dorsal horn neurons, basal cell carcinoma signaling, human embryonic

stem cell pluripotency, and molecular mechanisms of cancer. We found that many genes related to Wnt signaling (APC2, FZD5, WNT1, 2B, 3A, 5A, 5B, 5B, 6, 7B, 9B, 10A, and 10B) were involved in the identified pathways (Supplementary Table 1). To further investigate CpG methylation of the Wnt gene family, we charted the CpG methylation scores of each probe. In particular, WNT3A and WNT5A, which are key regulators of early development[17,18], exhibited increased gene body methylation in SALL3 KD cells compared to that in control cells (Fig. 6c). To elucidate the mechanisms by which SALL3 inhibits DNMT3B-mediated gene body methylation, we performed chromatin immunoprecipitation (ChIP)-seq analysis with SALL3 KD cells using anti-SALL3 and

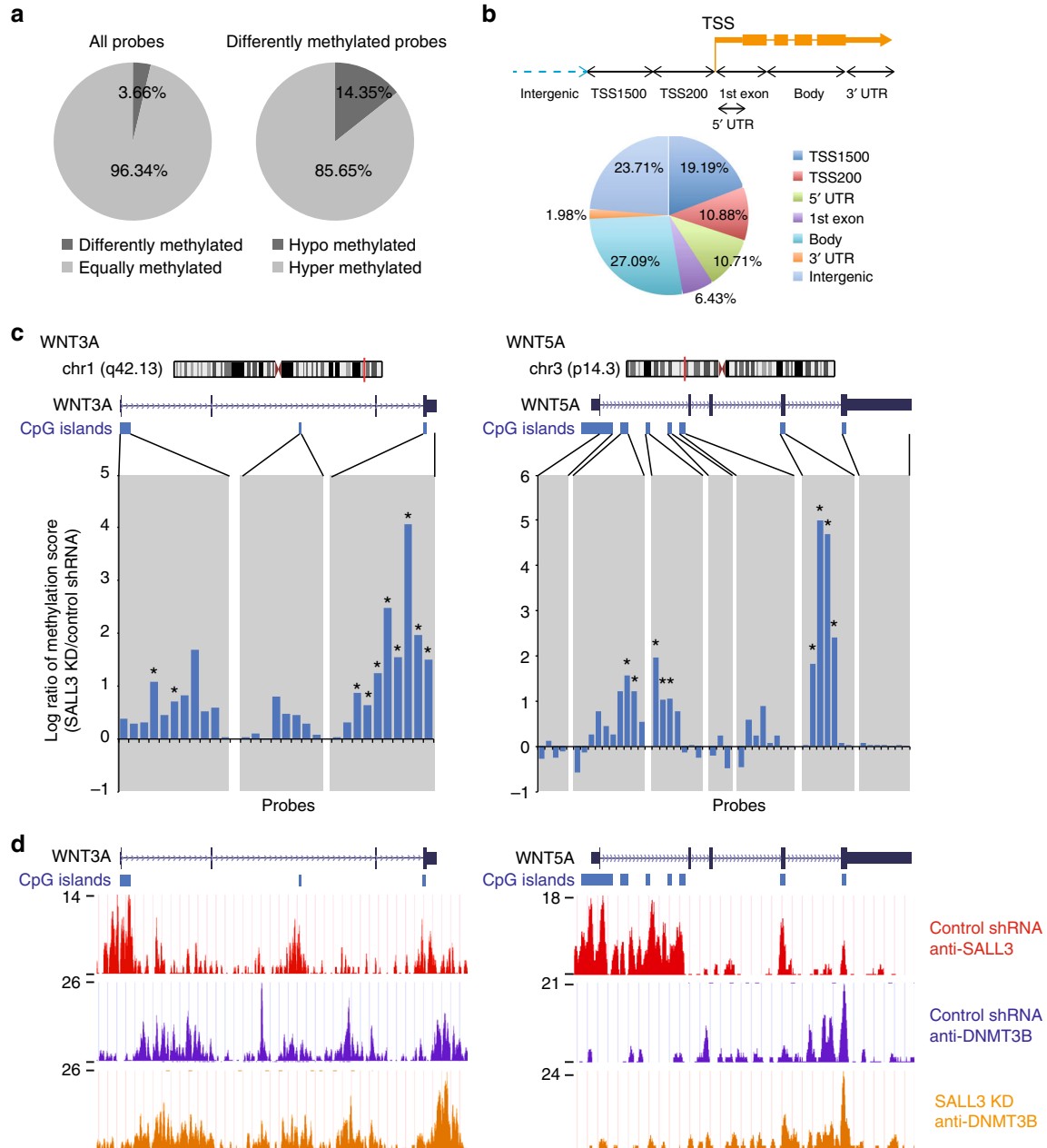

**Fig. 6** SALL3 regulates gene-body DNA methylation. **a** HumanMethylation450K BeadChip analysis of 253G1 *SALL3* shRNA cells and 253G1 control shRNA cells. Diagram shows a comparison of the similarly methylated and differentially methylated probes (left) and hypermethylated and hypomethylated probes in 253G1 *SALL3* shRNA cells, compared to the control cells (right). **b** Distribution of hypermethylated regions. Top: schematic showing the seven categories. Bottom: pie chart showing the percent distribution of hypermethylated probes in each category. **c** HumanMethylation450K BeadChip analysis of the WNT3A locus (left) and WNT5A locus (right). Top: schematic of a gene locus on the chromosome. Middle: schematic of a CpG island locus in two genes depicted by the UCSC Genome Browser. Bottom: methylation score. Gray vertical bars highlight CpG island and shore regions (*n* = 3, biological replicates). Asterisks highlight probes with a significant difference in the methylation score between the control and *SALL3* knockdown (*P* < 0.05, two-sided *t* test). **d** The ChIP-seq data of the WNT3A locus (left) and WNT5A locus (right) are depicted by the UCSC Genome Browser. Top track: SALL3 protein binding to genomic regions in 253G1 control shRNA cells. Middle and bottom tracks: DNMT3B protein binding to genomic regions in 253G1 control shRNA cells and 253G1 *SALL3* shRNA cells

anti-DNMT3B antibodies. SALL3 was found to be broadly enriched at CpG islands on genome including *WNT3A* and *WNT5A* genes of the control cells (Fig. 6d, top track). We found that 83.3% of SALL3-enriched intervals (see Methods) were localized within CpG islands (Supplementary Figure 8). The binding of DNMT3B to *WNT3A* and *WNT5A* genes was not restricted at CpG islands, regardless of *SALL3* KD (Fig. 6d, middle and bottom tracks). The ChIP-seq analysis also indicated that there was little

difference in the total DNMT3B binding distributions across gene bodies and promotor regions between the control and *SALL3* KD cells (Supplementary Figure 9). Importantly, however, we found that SALL3 proactively binds to CpG islands in gene bodies of *WNT3A* and *WNT5A* (Fig. 6d, top track), at which *SALL3* KD also led to aberrant DNA hypermethylation (Fig. 6c). In addition, the *SALL3* KD enhanced recruitment of DNMT3B to a part of CpG island and shore regions in these two genes, which was

associated with hypermethylation (Fig. 6d, middle and bottom tracks). *SALL3* KD also upregulated the binding of DNMT3B and DNA methylation at some of CpG island regions in gene bodies of *BMP4* and *DHH*, which are known to be involved in embryonic development[19] (Supplementary Figure 10). These results support that SALL3 plays critical roles in regulation of DNMT3B function and contributes to epigenetic changes in central genes involved in hiPSC differentiation.

## Discussion

In this study, we identified *SALL3* as a key marker that predicts the differentiation behavior of hiPSC lines toward three germ layers. Loss- and gain-of-function studies revealed that SALL3 regulates hiPSC differentiation into cardiac and neural cell lineages. Furthermore, our results suggest that SALL3 binds to and regulates DNMT3B in hiPSCs, and that downregulation of SALL3 increases CpG methylation in the gene bodies of Wnt signaling-related genes. To the best of our knowledge, we are the first to demonstrate that SALL3 plays a role as a switch of the differentiation propensity of hiPSCs toward different germ layer lineages.

Using ten commonly available hiPSC lines, we identified candidate genes to predict differentiation propensity into three germ layers with a rank correlation method. As the rank of the hiPSC lines in terms of differentiation capacity into ectoderm was inverse correlated with that into mesoderm, our candidate genes appear to be predictive markers for differentiation propensity but not for defects in differentiation per se. Koyanagi-Aoi et al.[4] reported that human endogenous retrovirus type-H (HERV-H) is specifically activated in hiPSC lines defective in differentiation into neural lineage cells. We did not identify genes regulated by LTR7 (type 7 long terminal repeat nucleic acid sequences) of HERV-H (*HHLA1*, *ABHD12B*, and *C4orf51*) among our candidate genes showing a significant association with ectoderm differentiation (Supplementary Data 3), which suggests that the neural differentiation defects by LTR7/HERV-H are limited in the hiPSC lines that we employed. Among our candidate genes, we identified *SALL3* as a relevant marker for predicting the differentiation propensity of hiPSC lines, playing a pivotal role as a switch to determine differentiation into the ectoderm and mesoderm lineages. We investigated the role of SALL3 in the nondirected, cardiac, and neural differentiation of hiPSCs. Although we attempted to directly differentiate *SALL3* KD cells into hepatocytes, an endoderm lineage, using a previously reported protocol[5], marked detachment of the cells from culture plates occurred in the late phase of hepatic differentiation for unknown reasons. Therefore, the suitability of *SALL3* as a marker of differentiation propensity into endoderm lineages has been confirmed only by nondirected EB formation.

*SALL3* is a member of the SALL gene family, which encodes a group of putative developmental transcription factors, and it has been highly preserved throughout evolution[20]. *Sall3* knockout mice exhibit plate deficiency, abnormalities in cranial nerves, and perinatal lethality[21], but little is known about the molecular mechanisms by which it controls embryonic development. A previous study showed that SALL3 physically interacts with the PWWP domain of DNMT3A rather than DNMT3B and reduces CpG island methylation by inhibiting the binding of DNMT3A to chromatin in hepatocellular carcinoma[14]. In contrast, our data indicated that SALL3 binds to DNMT3B, and the interaction with DNMT3A was not confirmed. This discrepancy may be due to the type of cells used in each study, as expression of DNMT3B predominates in hPSCs[15]. In addition to the selective interaction with DNMT3B, SALL3 was found to bind to CpG islands in gene bodies of *WNT3A* and *WNT5A* (Fig. 6d, top track), repressing

their methylation (Fig. 6c) as well as DNMT activity (Figs. 5c and 6a) in hiPSCs. We also found that the substitution of DNMT3B with a lower amount of DNMT3B with mutations, which do not reside within/in the vicinity of its functional domains[22], partially rescued the SALL3 deficiency-induced changes in lineage differentiation propensity (Fig. 5d, e). These results suggest that DMNT3B is functionally involved in the regulation of differentiation propensity by SALL3. Consistent with our observation, Martins-Taylor et al.[23] reported that the neural differentiation is accelerated in *DNMT3B* KD hESCs. Although the repression of DNMT3B function by SALL3 can explain the regulation of DNA methylation in hiPSCs, the detailed mechanisms of the effect of SALL3 on DNMT3B are to be elucidated. The levels of DNMT3B bound to a part of CpG island regions in gene bodies of several representative genes as well as DNMT activity was found altered by the *SALL3* KD, suggesting that conformational change, complex formation and/or recruitment blocking of DNMT3B by SALL3 could be one of the mechanisms. Indeed, DNMT3 family is proposed to form hetero- and homo-oligomers to stimulate their enzymatic activity[24].

Relating to the inhibitory role of SALL3 in DNMT3B function, *SALL3* KD influenced approximately 4% of DNA methylation sites in hiPSCs, which is consistent with the observation in *DNMT3B* knockout hESCs[15]. As noted above, we discovered that *SALL3* KD increased gene body methylation, especially that of Wnt family genes. A recent study by Baubec et al.[16] showed that DNMT3B1 recruitment and de novo gene body methylation of active genes scaled with the cotranscriptional deposition of H3K36me3. It has been suggested that gene body methylation could function similarly to H3K36me3 to promote proper splicing and chromatin compaction at active genes[16,24,25]. Despite the induction of hypermethylation in the gene bodies of Wnt family genes, SALL3 depletion resulted in no significant effects on the mRNA expression of Wnt family genes in undifferentiated hiPSCs (Supplementary Data 5). Gene body methylation in Wnt family genes may thus contribute to changes in gene expression at an optimal time point after the initiation of differentiation, rather than in the undifferentiated state. In fact, endogenous Wnt/β-catenin signaling is inactive in undifferentiated hESCs and active in differentiated hESCs[26].

In conclusion, our study offers an approach for identifying markers in the pluripotent state that are predictive of the EB differentiation behaviors of hiPSC lines using rank correlation. This approach could be used to identify differentiation prediction markers for other types of cells in hPSCs. In addition to the rapid screening of hPSC lines used as raw materials for CTPs, the expected advantages of appropriately selecting an hPSC line that is likely to differentiate into a desired cell type are as follows: (i) reduction in the contamination of CTPs with undesired cells; (ii) reduction in residual undifferentiated cells resistant to expected differentiation in CTPs; (iii) improvement in the final yield of desired cells in CTPs; and (iv) reduction in the production period for CTPs. Our approach should enable improved selection of hPSC lines suitable for future CTPs, and our study presents an entry point for defining the molecular mechanisms underlying hPSC differentiation.

## Methods

**Cell culture**. HiPSC lines were obtained from RIKEN Cell Bank (201B7, 253G1, 409B2, 606A1, 648A1, HiPS-RIKEN-1A, HiPS-RIKEN-2A, and HiPS-RIKEN-12A), the American Type Culture Collection (ATCC-DYR0100 hiPSC and ATCC-HYR0103 hiPSC), the JCRB Cell Bank (Tic), and System Biosciences (human mc-iPS). Detailed information about the hiPSC lines is described in Supplementary Table 2. Undifferentiated hiPSCs were maintained on Matrigel (Corning)-coated dishes in mTeSR1 medium (StemCell Technologies) or on SNL feeder cells (a mouse fibroblast STO cell line expressing the neomycin-resistance gene cassette and LIF) in human ES cell culture medium (ReproCell) supplemented with 4 ng

mL$^{-1}$ human basic fibroblast growth factor (bFGF; WAKO). All cells were cultured at 37 °C in a humidified atmosphere of 5% CO$_2$ and 95% air.

**mRNA microarray and data analysis.** Total RNA was isolated from hiPSC lines using an RNeasy Mini Kit (Qiagen) and treated with DNase I according to the manufacturer's instructions. RNA samples ($n = 6$, biological replicates) were converted into biotinylated cRNA using Two-Cycle Target Labeling and Control Reagents (Affymetrix). Labeled RNA was processed for microarray hybridization to Human Genome U133 Plus 2.0 GeneChips (Affymetrix). An Affymetrix GeneChip Fluidics Station was used to perform streptavidin/phycoerythrin staining. The hybridization signals on the microarray were scanned using a GeneChip Scanner 3000 (Affymetrix) and analyzed using Expression console software (Affymetrix). Normalization was performed by global scaling with the arrays scaled to a trimmed average intensity of 500 after excluding the 2% of probe sets with the highest and lowest values. The hybridization experiments were performed with six samples of each hiPSC line. To extract the informationally significant probe sets from the data, we filtered probe sets using the following three steps. First, probe sets were regarded as "present" when indicated as "present" by "absolute analysis" using Expression console software in four of the six samples from one cell line. Second, when no significant difference was observed among cell lines using one-way ANOVA ($P < 0.05$), probe sets were eliminated from the data set. Third, if the difference between the maximum and minimum mean values of the probe set in all strains was ≥fivefold, probe sets were used for analysis. A hierarchical clustering analysis was performed using R statistics software (R 3.3.2).

**Nondirected EB differentiation.** Undifferentiated hiPSCs were pre-cultured in human ES cell culture medium on feeder cells for 5–6 days. Before harvest, cells were cultured in EB medium [KO-DMEM (Invitrogen) supplemented with 20% KnockOut Serum Replacement (KSR, Invitrogen), 100 μM nonessential amino acids (Invitrogen), 2 mM L-glutamine (Invitrogen), and 55 μM 2-mercaptoethanol (Invitrogen)] with 4 ng mL$^{-1}$ bFGF for 1 day. Cells were treated with CTK solution (ReproCell) to eliminate dissociated feeder cells, and collected with a cell scraper. Cell clumps were plated on six-well low-adherence plates (Corning) in EB medium for 16 days, and the medium was refreshed every 2–3 days.

**Low-density PCR array and PCA.** Total RNA was isolated from EBs ($n = 6$, biological replicates) using an RNeasy Mini Kit (Qiagen) and treated with DNase I according to the manufacturer's instructions. Synthesis of cDNA was performed using the High Capacity RNA-to-cDNA Kit (Qiagen) with 1 μg total RNA per the manufacturer's instructions. A TaqMan array card (Applied Biosystems) containing primers and probes was dried in a 384-well plate. We curated sets of marker genes for each of the three germ layers (Supplementary Data 2). TaqMan 2× Universal PCR master mix (Applied Biosystems) was used for PCR analysis, which was carried out on an Applied Biosystems 7900HT thermal cycler under the following cycling conditions: incubation at 50 °C for 2 min and 10 min of denaturation at 94.5 °C, followed by 40 cycles of 97 °C for 30 s and 60 °C for 1 min. Each gene was run in duplicate. Gene-expression levels were calculated using the comparative Ct method with GAPDH (glyceraldehyde-3-phosphate dehydrogenase) as the endogenous housekeeping gene, and results were normalized to the average of the 201B7 samples to obtain fold changes. After data standardization (z-scoring) of each of the three germ layer marker genes, PCA was performed using SYSTAT 13 Software (Systat Software Inc.). The first principal component score (PC1) was calculated for further analysis.

**Rank correlation analysis.** To identify microarray probe sets related to the differentiation of hiPSCs into the three germ layers, correlations between the intensity value rank of the filtered probe sets and the PC1 rank in the 10 hiPSC lines was determined by calculating Spearman's rank correlation coefficients[27]. Probe sets exhibiting statistically significant correlations ($P < 0.05$) were selected. When there are $n = 10$ data points, the observed value of $r_s$ must be greater than 0.648 (positively correlated) or less than −0.648 (negatively correlated) to be considered statistically significant ($P < 0.05$)[28].

**Quantitative real-time polymerase chain reaction.** Total RNA was isolated as described above. qRT-PCR was performed with the QuantiTect Probe One-Step RT-PCR Kit (Qiagen) on a StepOnePlus Real-Time PCR System (Applied Biosystems). The expression levels of target genes were normalized to those of the GAPDH transcript or 18S rRNA, which were quantified using TaqMan human GAPDH control reagents (Applied Biosystems) or eukaryotic 18S rRNA endogenous control (Applied Biosystems), respectively. Probes and primers were obtained from Sigma-Aldrich. The sequences of the primers and probes used in the present study are listed in Supplementary Table 3.

**Generation of SALL3 KD cell lines.** SALL3 KD cells were generating by infecting 253G1 and 201B7 cells with lentiviral particles expressing SALL3-targeted shRNAs. Briefly, Lenti-X 293T cells were transfected with individual clones from a Sigma MISSION shRNA targeting set (No. 1: TRCN0000019754, No. 2: TRCN0000417790) or control shRNA plasmid along with a MISSION Lentivirus

Packaging Mix (Sigma-Aldrich) according to the manufacturer's instructions. Media containing viruses were collected 48 h after transfection, and cells were transduced with the viruses in the presence of 8 μg mL$^{-1}$ polybrene (Sigma-Aldrich) for 24 h. The cells were then subjected to selection by 2 μg mL$^{-1}$ puromycin (Gibco) for 48 h. The mRNA microarray data of the control and SALL3 shRNA cells were obtained as described above.

**Generation of a SALL3-overexpressing cell line.** SALL3 overexpression cells were generating by infecting 253G1 cells with lentiviral particles expressing SALL3. Briefly, the nucleotide sequence of the human SALL3 open reading frame (NM_171999) was de novo synthesized (Gen Script) and cloned into pLVSIN-EF1α puromycin vector (Takara Clontech). Packaging of lentivirus and infection of virus were performed as described above.

**Cardiomyocyte differentiation.** Differentiation of hiPSCs into cardiomyocytes was induced as previously reported[11], with a few modifications. hiPSCs were detached by incubation with StemPro Accutase (Thermo Fisher Scientific) for 7 min and seeded onto six-well cell culture plates (BD) coated with Matrigel at a density of 50,000 cells cm$^{-2}$ in mTeSR1 for 4–5 days before cardiomyocyte induction. Confluent hiPSCs were treated with 12 μM CHIR99021 (Stemgent) in RPMI1640 medium (Sigma-Aldrich) supplemented with B27 minus insulin (RPMI/B27-insulin; Invitrogen) for 24 h. On day 1, the medium was changed to RPMI/B27-insulin. On day 3, cells were treated with 5 μM IWP4 (Stemgent) in RPMI/B27-insulin for 48 h. On day 5, the medium was changed to RPMI/B27-insulin. The medium was changed to RPMI/B27 on day 7 and was then changed every 3 days. Differentiated cells were harvested at the indicated times for further analysis.

**Flow cytometry.** Cardiac differentiated hiPSCs were fixed using the BD Cytofix fixation buffer (BD Biosciences) for 20 min and permeabilized using BD Perm/Wash buffer (BD Biosciences) for 10 min at room temperature. The cells were incubated for 1 h at room temperature with mouse anti-cardiac troponin T monoclonal antibody (ab8295, Abcam). Indirect immunostaining was then completed with anti-mouse IgG Alexa Fluor 488-conjugated secondary antibody (A28175, Thermo Fisher Scientific) for 1 h. Normal mouse IgG antibody was used as a negative control (5415, Cell Signaling Technology). Stained cells were analyzed using an S3 cell sorter (BioRad). For flow cytometric analysis, live cells were gated using a SSC-area and FSC-area gate. Data retrieved from sorting was analyzed using FlowJo software (Tree Star).

**Neural differentiation.** Differentiation of hiPSCs into neural cells was induced as previously reported[12], with a few modifications. hiPSCs were detached by incubation with StemPro Accutase (Thermo Fisher Scientific) for 7 min and seeded onto six-well cell culture plates (BD) coated with Matrigel at a density of 50,000 cells cm$^{-2}$. Confluent hiPSCs were treated with 10 μM of the ALK inhibitor SB431542 (Stemgent) and 500 ng mL$^{-1}$ of Noggin (R&D) in DMEM/F12 medium containing 20% KSR. The medium was replaced on days 1 and 2. On day 5 of differentiation, SB431542 was withdrawn and increasing amounts of N2 media (25%, 50%, and 75%) were added to the knockout serum replacement medium every 2 days while maintaining 500 ng mL$^{-1}$ of Noggin.

**Hepatocyte differentiation.** Differentiation of hiPSCs into hepatocytes was induced as previously reported[5], with a few modifications. hiPSCs were detached by incubation with StemPro Accutase (Thermo Fisher Scientific) for 7 min and seeded onto six-well cell culture plates (BD) coated with Matrigel at a density of 20,000 cells cm$^{-2}$. Confluent hiPSCs were treated with 100 ng mL$^{-1}$ of activin A (R&D), 50 ng mL$^{-1}$ of Wnt3a (R&D), and 1 mM sodium butylate (Thermo Fisher Scientific) in RPMI1640 medium (Thermo Fisher Scientific) containing 1 × B27 (Thermo Fisher Scientific) for 24 h. On day 1, the sodium butylate was omitted from the medium. On day 3 of differentiation, the culture medium was replaced with knockout-DMEM containing 20% (vol/vol) KSR, 1 mM L-glutamine, 1% (vol/vol) nonessential amino acids, 0.1 mM 2-mercaptoethanol (all from Thermo Fisher Scientific), and 1% (vol/vol) DMSO (Sigma) for 7 days. Finally, the cells were cultured in hepatocyte culture medium (Lonza) supplemented with 20 ng mL$^{-1}$ HGF (PeproTech) and 20 ng mL$^{-1}$ oncostatin M (PeproTech) for another 7 days. The medium was changed daily during the differentiation period.

**Immunofluorescence staining.** Neural differentiated hiPSCs were fixed with 4% paraformaldehyde in PBS (Nacalai) for 20 min at room temperature. After washing with PBS, the cells were permeabilized with 0.2% Triton-X100 in PBS for 15 min and blocked with Blocking One (Nacalai) for 30 min. Samples were incubated for 1 h with rabbit anti-PAX6 (BioLegend PRB-278P-100, 1:200) and mouse anti-Oct-3/4 (BD 611202, 1:200) antibody. Indirect immunostaining was then completed with anti-mouse IgG/Alexa Fluor 488 (Thermo Fisher Scientific A28175, 1:1000) and anti-rabbit IgG/Alexa Fluor 555 (Thermo Fisher Scientific A27039, 1:1000) for 1 h and examined under a BZ-X710 fluorescence microscope (Keyence).

**Immunoprecipitation.** Cells were lysed for 30 min at 4 °C in lysis buffer (20 mM Tris-HCl, pH 8.0, 150 mM NaCl, 1% NP-40) containing complete mini EDTA-free protease inhibitor cocktail (Roche). Cell lysates were clarified by centrifugation for 30 min at 20,000 × g at 4 °C. Cleared lysates were then incubated with antibody for 2 h at 4 °C with rotation. The following antibodies were used: anti-SALL3 (Abnova PAB28233, 1:100), anti-DNMT3B (R&D Systems AF7646, 1:100), rabbit IgG (Cell Signaling Technology 2729, 1:100), and sheep IgG (R&D Systems 5-100-A, 1:100). The lysates were then incubated with protein G magnetic beads (Life Technologies) with rotation for 30 min at 4 °C. The beads were washed three times with lysis buffer. Proteins bound to the beads were eluted using sodium dodecyl sulfate (SDS) sample buffer solution with reducing reagent (Nacalai) at 95 °C for 5 min.

**Western blot analysis.** Cell lysates and immunoprecipitants were used for western blot analysis. Proteins were separated by SDS polyacrylamide gel electrophoresis, transferred to polyvinylidene fluoride membranes (Bio-Rad), and probed with primary antibody. The following antibodies were used: anti-SALL3 (Abnova PAB28233, 1:1000), anti-DNMT3A (Abnova 64B1446, 1:1000), anti-DNMT3B (R&D Systems AF7646, 1:1000), anti-LSD1 (Cell Signaling Technology 2184, 1:1000), and anti-β-actin (Sigma-Aldrich A5441, 1:1000). The membranes were incubated with horseradish peroxidase-conjugated anti-rabbit IgG (GE Healthcare, 1:5000), anti-mouse IgG (Invitrogen, 1:5000), or anti-sheep IgG (Abcam, 1:5000). Proteins were visualized with ECL Prime Western Blotting Detection Reagent (GE Healthcare) and ChemiDoc Touch Imaging System (BioRad). Uncropped images of scanned blots shown in Supplementary Figure 12 are provided in Supplementary Information file.

**Generation of a *SALL3*-knockout cell line.** The cells were culture-adapted to the Cellartis DEF-CS Culture System (Takara Clontech) prior to single-cell cloning according to the manufacturer's recommendations. Totally, 253G1 cells (1 × 10⁶ cells) were transfected with 7.5 µg CRISPR-Cas9 plasmid (pCMV-Cas9-GFP, Sigma-Aldrich) using a NEPA21 electroporator (Nepa Gene). The CRISPR-Cas9 plasmid carried the target sequence 5′-CGTCCGACTTGAGGTGCTGGGG-3′ in the guide RNA gene (Target ID: HS0000537218). After 12 days of transfection, DNA double-strand-break activity of the CRISPR-Cas9 was confirmed with CEL-I assay kit (Transgenomic) using genomic DNA extracted from an aliquot of transfected cells (F-primer: 5′-ATGGCCATGCATTATTCACC-3′, R-primer: 5′-CAGGAAGCAGGGAACTTTCT-3′). The transfected cells were seeded in 96-well plates at a very low density in the Cellartis iPSC Single-Cell Cloning DEF-CS Culture Media (Takara Clontech) according to the manufacturer's recommendations. Individual colonies derived from single cell were picked and expanded, and then genomic DNA was purified using NucleoSpin Tissue XS (Macherey-Nagel) according to the manufacturer's instructions. To screen these clones, we performed qPCR with the genomic DNA using probe and primers that targeted to the mutation site (F-primer: 5′-GCATGTCTCGGCGCAAGC-3′, R-primer: 5′-CCTC ACCGTGCTCGGGAG-3′, probe: 5′-FAM-AAGCCCCAGCACCTCAAGTCGGA CG-BHQ1-3′). The clones exhibiting no signal of PCR amplification were selected as candidates for *SALL3* knockout cells. Finally, homozygous mutation was confirmed by Sanger sequencing and western blotting (Supplementary Figure 10).

**Generation of a *DNMT3B*-mutant cell line.** The *SALL3⁻/⁻DNMT3B⁻/mut* cells were generating by infecting 253G1 *SALL3⁻/⁻* cells with lentiviral particles expressing *DNMT3B*-targeted CRISPR/Cas9 system (Vector: pLV-U6g-EPCG, Sigma-Aldrich). The cells were culture-adapted to the Cellartis DEF-CS Culture System (Takara Clontech) prior to infection according to the manufacturer's recommendations. The CRISPR-Cas9 plasmid carried the target sequence 5′-TCCTCTCCATTGAGATGCCTGG-3′ in the guide RNA gene (Target ID: HS0000200662). Cells were transduced with the viruses in the presence of 8 µg/mL polybrene (Sigma-Aldrich) for 24 h. The cells were then subjected to selection by 2 µg/mL puromycin (Gibco) for 48 h. CEL-I assay (F-primer: 5′-CTAAGAATGCA TCCTGGGGC-3′, R-primer: 5′-GCTCACTATGTCAGCGCCT-3′) and single-cell cloning were performed as described above. To screen these clones, we performed qPCR with the genomic DNA using probe and primers that targeted to the mutation site (F-primer: 5′-TCCCTGCTTCCCTTTCACCC-3′, R-primer: 5′-CGTTGACGAGGATCGAGTCTTC-3′, probe: 5′-FAM-CGTCCTCCTCTCCATTGAGATGCCTGGTG-TAMRA1-3′). The clones exhibiting no signal of PCR amplification were selected as candidates for *DNMT3B*-mutant cells. Finally, homozygous mutation was confirmed by Sanger sequencing and western blotting (Supplementary Figure 7).

**Methyl array.** We used the Illumina Infinium DNA methylation platform's HumanMethylation450 (HM450) BeadChip (Illumina) to obtain the gene promoter and gene body DNA methylation profiles of *SALL3* KD 253G1 cells and control shRNA 253G1 cells. For each sample, 0.5 µg of DNA was bisulfite-converted using the EZ DNA Methylation Kit (Zymo Research) and analyzed on an HM450 BeadChip according to the manufacturer's instructions. The Illumina GenomeStudio program was used for normalization and extraction of the methylated and unmethylated signal intensities. Probe characteristics were derived from IHumanMethylation450_15017482_v1.2.bpm (peak-based correction method)[29]. For all samples, detection *P* values were <0.01 for over 99.9 % of all loci;

nonsignificant loci (*P* ≥ 0.01) were not considered in the subsequent analysis. Then, probes without annotations (chromosome/locus) were removed, yielding a final data set consisting of 485,529 (out of 485,577) assessed loci. Statistical analysis was performed with *t* tests to compare control shRNA 253G1 cells (*n* = 3) and *SALL3* KD 253G1 cells (*n* = 3). The obtained *P*-values were corrected by the Benjamini–Hochberg method, and *q* values were calculated. Probes satisfying *q* < 0.05 and with an intergroup variation ratio of |log₂ (fold-change)| > 0.6 were determined to exhibit significant differences. For the HM450 DNA methylation array, quantitative scores of DNA methylation levels were obtained as b-values determined from the Illumina analysis, ranging from "0" for completely unmethylated to "1" for completely methylated.

**DNMT activity assay.** Nuclei were isolated from the *SALL3* knockout cells and wild-type 253G1 hiPSCs with the 8-min Cytoplasmic & Nuclear Protein Extraction Kit (101Bio). DNMT activity of the nuclear fractions (5 µg protein per sample) was measured with the EpiQuik DNMT Activity/Inhibition Assay Ultra Kit (Epigentek) according to the manufacturer's instructions. Protein concentrations were determined with the Pierce BCA protein assay kit (Thermo Fisher Scientific).

**ChIP-seq.** *SALL3* KD (253G1 shRNA-*SALL3*) cells and control shRNA (253G1 control-shRNA) cells were fixed and sent to Active Motif for ChIP, library preparation, sequencing, and initial data analysis. Rabbit polyclonal anti-DNMT3B antibody (sc-20704, Santa Cruz Biotechnology) and anti-SALL3 antibody (PAB28233, Abnova) was used for ChIP[30]. Sequencing was carried out with 75-bp reads on the NextSeq 500 platform (Illumina). Since the 5′-ends of the aligned reads (tags) represent the ends of the ChIP/IP-fragments, the tags were extended in silico (using Active Motif software) at their 3′-ends to a length of 150–250 bp, depending on the average fragment length in the size-selected library (normally 200 bp). To identify the density of fragments (extended tags) along the genome, the genome was divided into 32-nt bins, and the number of fragments in each bin was determined. The generic term "interval" was used to describe genomic regions with local enrichments in tag numbers. Intervals are defined by the chromosome number and a start and end coordinate. The two main peak callers used at Active Motif are MACS[30] and SICER[31]. To compare peak metrics between two or more samples, overlapping intervals are grouped into "active regions", which are defined by the start coordinate of the most upstream interval and the end coordinate of the most downstream interval (the union of overlapping intervals, e.g., "merged peaks"). After defining the intervals and active regions, their genomic locations along with their proximities to gene annotations and other genomic features were determined.

**Statistical analysis.** Statistical analysis was performed using SigmaPlot 12.5 software (Systat Software Inc.). Statistical comparisons were made using Student's *t* tests or one-way ANOVA with post hoc Tukey–Kramer test. *P* values < 0.05 were considered significant.

**Reporting Summary.** Further information on experimental design is available in the Nature Research Reporting Summary linked to this article.

## Data availability

mRNA microarray and ChIP-seq data sets generated for this study are available from the NCBI-GEO database under Accession numbers GSE88963, GSE114977, and GSE104863. The authors declare that all the other data supporting our findings are available within the article and its Supplementary files and from the corresponding author upon reasonable request.

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

## Acknowledgements

The authors thank the Center for iPSC Research and Application, Kyoto University, for the 201B7, 253G1, 409B2, 606A1, and 648A1 cell lines. The present work was supported by research grants from the Japanese Ministry of Health, Labor and Welfare (Marketing Authorization Facilitation Program for Innovative Therapeutic Products) and the Japan Agency for Medical Research and Development (15mk0104064h0101, 15mk0104064h0301, 16mk0104044j0002, 16mk01040044j0202, 17mk0104044j0003, and 17mk0104044j70203).

## Author contributions

T.K., S.Y., and Y.S. conceived and designed the research. T.K., S.Y., and S.M. performed the experiments. T.K., S.Y., S.K., K.T., T.M., A.M., and Y.S. analyzed the data. T.K., S.Y., and Y.S. wrote the papermanuscript with input from the other authors. S.Y., A.M., and Y.S. acquired the funding.

## Additional information

**Competing interests:** The authors declare no competing interests.

