## [Peer Review File · Nature Communications]

Reviewers' comments:

Reviewer #1 (Remarks to the Author):

In this study entitled 'SALL3 regulates propensity of human induced pluripotent stem cells to differentiate into distinct cell lineages', Kuroda et al show that SALL3 predicts differentiation potential of human iPS cell lines by testing 10 different human iPS cell lines. The authors claim that the level of SALL3 in the cell correlates positively with ectoderm differentiation and negatively with mesoderm/endoderm differentiation during embryoid body formation. Similarly, the authors also show that SALL3 inversely regulates the capacities of cardiac and neural differentiation. Additionally, the authors present that SALL3 interacts with DNMT3B and mediates changes in DNA methylation pattern in human iPS cells. Since multiple prior studies suggested that human ES and iPS cell lines have variation in their potential during in vitro differentiation, careful characterization of individual human ES and iPS cell lines is critical for the future therapeutic applications in regenerative medicine. In this regard, the manuscript seems timely and important. However, there are multiple problems with data presented and analysis as described below. As a result, some conclusions of the manuscript are not strongly supported by the presented data.

Major comments:

1. Figure 2A. It is unclear how the authors obtain SALL3 as the only gene showing the inverse correlation of the lineage marker gene expression. Please describe the analysis methods in detail.
2. R-21A iPS cell line shows relatively high expression level of SALL3 in Figure 2B. However, this line does not show similar expression profiles for lineage marker genes to those of 201B7, 253G1, mc-iPS, and R-1A lines (Figure 1C). Surprisingly, the R-21A line behaves like other lines with lower SALL3 expression level such as Ai-1000, Ai-203, R2A, and R12A. Is this indicating that some iPS cell lines do not follow the main claim of the manuscript? It seems like that throughout the manuscript, the authors mainly test 253G1 line (and 201B7 line for some supplemental data). However, to generalize and strengthen the main claim of the manuscript, it will be important to test additional iPS cell lines.
3. The authors should test iPS cell lines with lower SALL3 expression level (Ai-1000, Ai-203, R2A, and R12A) by overexpression of SALL3. If the tested cell lines show enhanced ectoderm differentiation with impaired meso/endoderm differentiation upon overexpression of SALL3, the results will greatly strengthen the main claim of the manuscript.
4. Figure 5: Generally, the link between SALL3 functions and DNMT3B seems severely underdeveloped and does not explain the predicted roles of SALL3 and the propensity of iPS cells during differentiation. I do not understand why the authors present DNMT3B related data in the manuscript since the results do not make any conclusion.
5. SALL3 is a DNA binding protein. Chromosomal targets of SALL3 should be mapped to test if SALL3 directly regulates specific lineage marker genes.

Minor comments:

1. Abstract line 36: 'Without affecting pluripotency of hiPSCs, SALL3 knockdown inhibited ectoderm differentiation and conversely enhanced mesodermal/endodermal differentiation'. 'pluripotency' should be 'self-renewal' in the sentence.
2. Line 98: 'Using total RNA isolated from the EBs, we obtained the transcript expression profiles of 76 genes as three germ layer markers after 16 days of EB differentiation (Fig. S1)'. In Figure S1, expression profiles of >90 genes are shown and some of the genes are listed multiple times. Please correct the sentence if there is a mistake. It is also unclear why the results in Figure S1 were normalized to the average of the 201B7 samples to obtain fold changes. Please clarify.

Reviewer #2 (Remarks to the Author):

Human iPSCs have variety for differentiation capacity depending on clonal variation, which is big issue for clinical application. Kuroda et al., identified very good maker, SALL3, whose expression is closely correlate with the capacity of ectoderm differentiation. They also provided evidence that artificial changes of SALL3 expression in hiPSCs allow us to control the direction of differentiation from hiPSCs. This study indicated that SALL3 would be useful marker to select clinical-grade hiPSCs. However, this study didn't show the molecular mechanisms of SALL4 in ectoderm differentiation, therefore, the authors should consider the following points before a decision on its suitability for publication can be made.

Comment 1

This study identified SALL3 as a marker for ectoderm differentiation based on the correlation of expression level and differentiation capacity (Fig. 2A). However, because they didn't present detail information regarding the criteria for measurement of correlation. They should show the detail criteria for ectoderm positive correlation and mesoderm/endoderm negative correlation. Furthermore, they should show the gene lists for venn diagram in supplementary Table.

Comment 2

This study provided evidence that expression level of SALL3 in hiPSCs is critical for differentiation propensity. It should be important to clarify which period of SALL3 expression is critical for ectoderm differentiation. They should perform doxycycline inducible expression of SALL3 in SALL3 KD hiPSCs to investigate whether transient expression of SALL3 at pluripotent stage can rescue the differentiation defect of SALL3 KD hiPSCs.

Comment 3

Comparison of transcriptome data between wild type hiPSCs and SALL3 KD should be presented to clarify SALL3 function in ectoderm differentiation.

Comment 4

The authors identified hypermethylation regions by SALL3 KD hiPSCs. They should perform ChIP-seq of SALL3 in hiPSCs to identify the direct target and comparison between SALL3-binding region and hypermethylated regions.

Comment 5

This study showed that SALL3 interacts with DNMT3B and hypermethylation was observed at some locus by SALL3 KD in hiPSCs. Based on this observation, they claimed that abnormal hypermethylation by SALL3 KD is caused for abnormality of differentiation into ectoderm. However, they presented no data regarding molecular mechanism for the failure of differentiation into ectoderm by SALL3 KD. The authors should, at least, clarify whether the suppression of DNMT3B function by SALL3 is responsible for proper differentiation of hiPSCs into ectoderm. DNMT3B KD experiment in SALL3 KD hiPSCs should be performed. If DNMT3B KD can't rescue the phenotype of SALL3 KD, the authors should eliminate the data about interaction between SALL3 and DNMT3B and methylation analysis. Further, they should present additional data regarding molecular mechanism of SALL3 in ectoderm differentiation.

Reviewer #3 (Remarks to the Author):

The differentiation property of hiPSCs is critical for selection of clinical-grade stem cells. Kuroda et al.

profiled transcriptome of hiPSCs and differentiation markers in EBs, and found that each hiPSCs line has distinct differentiation biases. Authors found that the expression SALL3 is highly correlated with the neuroectodermal differentiation potential but negatively correlated with mesodermal differentiation potential of hiPSCs. Authors used overexpression and depletion of SALL3 to demonstrate the function of SALL3 in neuroectoderm differentiation. Their findings are interesting, but preliminary. The function of Sall3 in neuroectoderm development has been reported before (Parrish, et al, MCB, 2004), and function of SALL3 in inhibition of DNA methylation through binding with DNMT3A was investigated in hepatocellular carcinoma (Shikauchi, et al, 2009). Although the focus of current manuscript is to determine the marker for biased differentiation potential of hiPSCs, statistical power is limited. Despite the weakness of the manuscript, manuscript still presents important concept of characterizing the hiPSCs based on expression of a few selected markers, and describes the function of SALL3 in neuroectoderm development in human embryonic context. Below are comments to improve the manuscript.

Major comments

1. Using the 10 hiPSC lines to identify the SALL3 as a marker for higher neuroectodermal differentiation potential is impressive. However, R-12A in current study expresses high level of SALL3 (Figure 2B) but show less neural differentiation (Figure C). Thus, the expression level of SALL3 is not absolute determinant of neuroectodermal potential. Authors should perform meta analysis including other hiPSC or hESC studies that performed gene expression and comparison of differentiation potential to demonstrate that SALL3 expression implicates the biased neuroectoderm differentiation potential in a statistically significant manner. Studies by Koyanagi-Aoi, et al, PNAS, 2013; Salomonis, et al, Stem cell Reports, 2016; Nishiazawa, et al, Cell Stem Cell, 2016 among others should be analyzed to show the positive or negative correlation of SALL3 expression from neuroectoderm or mesoderm differentiation.

2. hiPSCs are highly heterogeneous and their transcriptome changes during serial passage, because of epigenetic memory (Kim et al., Nature, 2010), aberrant epigenetic DNA methylation (Weissbein et al., PLoS Genetics, 2017) and X chromosome erosion (Mekhoubad et al., 2012; Nazor et al., 2012)). Therefore, parental cell, exact passage number and gender are key factor to produce differentiation biases in hiPSCs. Authors need to describe and examine whether the differentiation biases that they observed are dependent on these factors or not.

3. Lineage commitment of cells in EB is highly dependent on their environment. In addition, authors evaluated only differentiation potential by in vitro differentiation. To validate whether their differentiation potential is not dependent on medium or in vitro conditions, they also should evaluate it by another assay, such as teratoma assay by overexpressing or knockdown SALL3 and injection.

4. Several groups reported that gene body DNA methylation plays active role in transcription (Ball et al., 2009). However, hypermethylation on gene body in Sall3 KD was observed in neuronal-related genes, which were downregulated in Sall3 KD. Authors should explain this contradiction. Authors should perform the correlation of gene body or promoter methylation affected by SALL3 knockdown, and perform validation experiments, such as qPCR.

5. Since their CHIP-seq does not show significant difference, SALL3 may regulate DNA methyltransferase activity of DNMT3B rather than DNA binding property. Therefore, I propose DNMT activity assay experiment in WT and SALL3-depleted cells to dissect SALL3-mediated inhibition of DNA methylation. Also, authors need to examine whether the binding sites of SALL3 is correlated with change in DNA methylation or DNMT3B binding sites.

Minor comments

1. Description of bioinformatics analysis is limited and it is hard to understand how authors performed the analysis. More detail is needed. For example, R software has multiple functions for hierarchical clustering (in 352 line in page 23). Authors need to describe which library they used and what parameters they used. Version information of software is also needed.

2. Heatmap requires "gradient color key" rather than simple color explanation. Authors also need to describe what heat colors represent (raw microarray intensity, z-score or any normalized values?) rather than "gene expression".

Response to reviewers:

Reviewer 1#

In this study entitled ‘SALL3 regulates propensity of human induced pluripotent stem cells to differentiate into distinct cell lineages’, Kuroda et al show that SALL3 predicts differentiation potential of human iPS cell lines by testing 10 different human iPS cell lines. The authors claim that the level of SALL3 in the cell correlates positively with ectoderm differentiation and negatively with mesoderm/endoderm differentiation during embryoid body formation. Similarly, the authors also show that SALL3 inversely regulates the capacities of cardiac and neural differentiation. Additionally, the authors present that SALL3 interacts with DNMT3B and mediates changes in DNA methylation pattern in human iPS cells. Since multiple prior studies suggested that human ES and iPS cell lines have variation in their potential during in vitro differentiation, careful characterization of individual human ES and iPS cell lines is critical for the future therapeutic applications in regenerative medicine. In this regard, the manuscript seems timely and important. However, there are multiple problems with data presented and analysis as described below. As a result, some conclusions of the manuscript are not strongly supported by the presented data.

Response: We thank the reviewer for the positive remarks.

Major comments:

1. Figure 2A. It is unclear how the authors obtain SALL3 as the only gene showing the inverse correlation of the lineage marker gene expression. Please describe the analysis methods in detail.

Response: We thank the reviewer for raising the point. In order to clarify the detail information regarding the criteria for identifying SALL3 correlated to differentiation propensity, we edited the results to explain the analysis method in detail (page 9, line 13---page 10, line 6), and corrected the venn diagrams in Figure 2A. Dataset S3 shows lists of candidate genes, of which expression was positively or negatively correlated to differentiation propensity toward three germ layers in hiPSC lines, and their Spearman's rank correlation coefficient (r_s) with a statistical significance. As the corrected venn diagrams shows, SALL3 appears to be the only gene satisfying both of a positive

(negative) correlation in ectoderm differentiation and a negative (positive) correlation in mesoderm/endoderm differentiation.

2. R-21A iPS cell line shows relatively high expression level of *SALL3* in Figure 2B. However, this line does not show similar expression profiles for lineage marker genes to those of 201B7, 253G1, mc-iPS, and R-1A lines (Figure 1C). Surprisingly, the R-21A line behaves like other lines with lower *SALL3* expression level such as Ai-1000, Ai-203, R2A, and R12A. Is this indicating that some iPS cell lines do not follow the main claim of the manuscript? It seems like that throughout the manuscript, the authors mainly test 253G1 line (and 201B7 line for some supplemental data). However, to generalize and strengthen the main claim of the manuscript, it will be important to test additional iPS cell lines.

Response: We agree with the reviewer's comment that some hiPSC line do not match its expression level of *SALL3* to their differentiation propensity. First, we would like to ascertain that *SALL3* gene was statistically identified with a rank correlation analysis of differentiation capability in hiPSC line. This indicates that behavior of all the hiPSC lines should be perfectly consistent with levels of *SALL3* expression only when the Spearman's rank correlation coefficient (r_s) is 1 or -1. Therefore, differentiation propensity of some hiPSC line may not follow the order of the levels in *SALL3* expression although the correlation remains statistical significant between differentiation propensity and *SALL3* expression among hiPSC lines. According to the reviewer's suggestion, to confirm our hypothesis that *SALL3* could be a marker for differentiation propensity, we additionally employed other two hiPSC lines, 606A1 and 648A1, as a test set and differentiated these hiPSCs into EBs. As expected from the *SALL3* expression levels in hiPSC lines, EBs derived from 606A1 hiPSCs exhibited higher expression of ectoderm marker genes, and lower expression of both mesoderm marker genes and endoderm marker genes, compared with 648A1 hiPSCs. These results are shown in the created Fig. S2.

3. The authors should test iPS cell lines with lower *SALL3* expression level (Ai-1000, Ai-203, R2A, and R12A) by overexpression of *SALL3*. If the tested cell lines show enhanced ectoderm differentiation with impaired meso/endoderm differentiation upon

overexpression of *SALL3*, the results will greatly strengthen the main claim of the manuscript.

Response: We thank the reviewer for the comment. To address the reviewer's comment, we established a *SALL3*-overexpressing cell line by stably introducing EF1alpha promoter-driven *SALL3* gene into R-2A cell line, which had shown the lowest *SALL3* expression and propensity for ectoderm differentiation among 10 hiPSC lines. We confirmed improvement of neural differentiation by overexpressing *SALL3* in a strain R-2A as expected. These results are shown in the created Fig. S6.

4. Figure 5: Generally, the link between *SALL3* functions and DNMT3B seems severely underdeveloped and does not explain the predicted roles of *SALL3* and the propensity of iPS cells during differentiation. I do not understand why the authors present DNMT3B related data in the manuscript since the results do not make any conclusion.

Response: We thank the reviewer for the critical comment. First, to reveal the function of *SALL3*, we established *SALL3* knockout cells and measured DNMT activity of the nuclear fraction prepared from the cells. The knockout of *SALL3* gene markedly increased endogenous DNMT activity of hiPSCs, suggesting the inhibitory role of *SALL3* in molecular function of DNMTs. Unfortunately, the quantitative measurement of the endogenous DNMT3B isoform activity after immunoprecipitation with anti-DNMT3B antibody was not successful. The procedures of establishment of *SALL3* knockout cell clones are described in the Materials and Methods section (page 30, line 11---page 31, line 15), and the results are described in the Results section (page 15, line 13---page 16, line 5) and shown in the created Fig. 5C. To clarify that *SALL3* selectively bind to DNMT3B but not other DNMTs contributing DNMT activity, we additionally demonstrate that *SALL3* failed to interact with a maintenance enzyme DNMT1 as well as DNMT3A in hiPSCs. These results are shown in the created Fig. 5A. In ChIP-seq analysis with anti-DNMT3B antibody, we could also find that *SALL3* KD enhanced recruitment of DNMT3B to some of CpG island regions of gene body in *WNT3A*, *WNT5A*, *BMP4* and *DHH* genes although there was little difference in the total DNMT3B binding distributions across gene bodies and promotor regions on genome between the control and *SALL3* KD cells. These results are shown in the created Fig. 5G

and Fig. S9. Taken together, we have concluded that SALL3 regulates DNMT3B and influence CpG methylation in gene body of genes involved in hiPSC differentiation.

5. SALL3 is a DNA binding protein. Chromosomal targets of SALL3 should be mapped to test if SALL3 directly regulates specific lineage marker genes.

Response: As the reviewer suggested, we firstly checked the DNA binding of SALL3 by ChIP-seq analysis. According to the ChIP-seq data, SALL3 was broadly enriched at CpG islands on genome including *WNT3A* and *WNT5A* genes. Importantly, SALL3 proactively bound to CpG islands in gene bodies of *WNT3A* and *WNT5A*, at which *SALL3* KD also led to aberrant DNA hypermethylation. These are described in the Results section (page 17, line 13---page 18, line 16) and shown in the created Fig. 5G and Fig. S7.

Minor comments:

1. Abstract line 36: ‘Without affecting pluripotency of hiPSCs, SALL3 knockdown inhibited ectoderm differentiation and conversely enhanced mesodermal/endodermal differentiation’. ‘pluripotency’ should be ‘self-renewal’ in the sentence.

Response: We thank the reviewer for pointing out this mistake. We have corrected accordingly.

2. Line 98: ‘Using total RNA isolated from the EBs, we obtained the transcript expression profiles of 76 genes as three germ layer markers after 16 days of EB differentiation (Fig. S1)’. In Figure S1, expression profiles of >90 genes are shown and some of the genes are listed multiple times. Please correct the sentence if there is a mistake. It is also unclear why the results in Figure S1 were normalized to the average of the 201B7 samples to obtain fold changes. Please clarify.

Response: We thank the reviewer for carefully reading the text. Genes used as markers for three germ layers in this study actually include ones, which were commonly reported for two or three germ layers. We have corrected the number of genes in the Results section as follows: total 97 genes comprised of 45 ectoderm markers, 56 mesoderm markers and 27 endoderm markers, which included common markers for two or three germ layers. The

reason for using the 201B7 sample as a reference is because the 201B7 strain is suitable for a standard cell line differentiating into all germ layer derivatives and is commonly used in many laboratories.

Reviewer 2#

Human iPSCs have variety for differentiation capacity depending on clonal variation, which is big issue for clinical application. Kuroda et al., identified very good marker, SALL3, whose expression is closely correlate with the capacity of ectoderm differentiation. They also provided evidence that artificial changes of SALL3 expression in hiPSCs allow us to control the direction of differentiation from hiPSCs. This study indicated that SALL3 would be useful marker to select clinical-grade hiPSCs. However, this study didn't show the molecular mechanisms of SALL4 in ectoderm differentiation, therefore, the authors should consider the following points before a decision on its suitability for publication can be made.

Response: We thank the reviewer for the positive remarks.

Comment 1

This study identified SALL3 as a marker for ectoderm differentiation based on the correlation of expression level and differentiation capacity (Fig. 2A). However, because they didn't present detail information regarding the criteria for measurement of correlation. They should show the detail criteria for ectoderm positive correlation and mesoderm/endoderm negative correlation. Furthermore, they should show the gene lists for venn diagram in supplementary Table.

Response: We thank the reviewer for raising the point. As we described in the response to the comment 1 of reviewer 1#, in order to clarify the detail information regarding the criteria for identifying SALL3 correlated to differentiation propensity, we edited the results to explain the analysis method in detail (page 9, line 13---page 10, line 6), and corrected the venn diagrams in Figure 2A. Dataset S3 shows lists of candidate genes, of which expression was positively or negatively correlated to differentiation propensity toward three germ layers in hiPSC lines, and their Spearman's rank correlation coefficient (rs) with a statistical significance. As the corrected venn diagrams shows, SALL3 is the only gene satisfying a positive (negative) correlation in ectoderm differentiation and a negative (positive) correlation in mesoderm/endoderm differentiation.

Comment 2

This study provided evidence that expression level of SALL3 in hiPSCs is critical for

differentiation propensity. It should be important to clarify which period of SALL3 expression is critical for ectoderm differentiation. They should perform doxycycline inducible expression of SALL3 in SALL3 KD hiPSCs to investigate whether transient expression of SALL3 at pluripotent stage can rescue the differentiation defect of SALL3 KD hiPSCs.

Response: We have previously blocked SALL3 expression by transiently transfecting siRNA oligonucleotides into hiPSCs and differentiated the transfected cells into ectoderm and mesoderm cell lineages. Although we confirmed the successful knockdown of SALL3 in hiPSCs, no significant effect on the differentiation propensity was observed. We presume that hiPSCs need a time enough to modify their epigenetic profiles under such a condition as persistent SALL3 depletion before induction of differentiation. Therefore, the experiments using doxycycline inducible expression of SALL3 would not be suitable for investigating the influence of SALL3 KD on differentiation in hiPSCs.

Comment 3

Comparison of transcriptome data between wild type hiPSCs and SALL3 KD should be presented to clarify SALL3 function in ectoderm differentiation.

Response: We thank the reviewer for the suggestion. We examined the comprehensive transcriptional profiles of wild type and SALL3 KD hiPSCs using microarray analysis. We were not able to find any difference in the mRNA expression of genes directly related with ectoderm differentiation between the undifferentiated control and SALL3 KD cells for the purpose of explaining ectoderm differentiation propensity of hiPSCs. Gene changes by SALL3 KD in hiPSCs are shown as the created Dataset S5. These results also suggest an importance of epigenetic changes in undifferentiated hiPSCs, of which gene expression would change accompanied with differentiation.

Comment 4

The authors identified hypermethylation regions by SALL3 KD hiPSCs. They should perform ChIP-seq of SALL3 in hiPSCs to identify the direct target and comparison between SALL3-binding region and hypermethylated regions.

Response: As described in response to the comment 5 of reviewer 1#, we checked the DNA binding of SALL3 by ChIP-seq analysis. According to the ChIP-seq data, SALL3 was broadly enriched at CpG islands on genome including *WNT3A* and *WNT5A* genes. Importantly, SALL3 proactively bound to CpG islands in gene bodies of *WNT3A* and *WNT5A*, at which *SALL3* KD also led to aberrant DNA hypermethylation. These are described in the Results section (page 17, line 13---page 18, line 16) and shown in the created Fig. 5G and Fig. S7.

Comment 5

This study showed that SALL3 interacts with DNMT3B and hypermethylation was observed at some locus by SALL3 KD in hiPSCs. Based on this observation, they claimed that abnormal hypermethylation by SALL3 KD is caused for abnormality of differentiation into ectoderm. However, they presented no data regarding molecular mechanism for the failure of differentiation into ectoderm by SALL3 KD. The authors should, at least, clarify whether the suppression of DNMT3B function by SALL3 is responsible for proper differentiation of hiPSCs into ectoderm. DNMT3B KD experiment in SALL3 KD hiPSCs should be performed. If DNMT3B KD can't rescue the phenotype of SALL3 KD, the authors should eliminate the data about interaction between SALL3 and DNMT3B and methylation analysis. Further, they should present additional data regarding molecular mechanism of SALL3 in ectoderm differentiation.

Response: We thank the reviewer for the suggestion and agree to an importance of rescue experiments by DNMT3B KD. In our manuscript, we referred the paper reported by Martins-Taylor et al., which showed that the timing of neural differentiation is altered in DNMT3B KD hESCs. Recently, Vera et al. have reported that DNMT3B knockout is able to partially revert the defect in neuronal differentiation induced by triple-knockout of TET enzymes (Nature Genet (2018) 50, 83-95). These suggest that depletion of only DNMT3B in hESCs can show a strong phenotype in neuronal differentiation under the severe condition such as complete block of DNA demethylation. Therefore, it would be difficult to rescue the phenotype of SALL3 KD cells by DNMT3B KD even though DNMT3B actually play critical roles in ectoderm differentiation in hPSCs. Further studies should be performed to elucidate the roles of DNMT3B in ectoderm differentiation of hPSCs in the future.

Reviewer 3#

The differentiation property of hiPSCs is critical for selection of clinical-grade stem cells. Kuroda et al. profiled transcriptome of hiPSCs and differentiation markers in EBs, and found that each hiPSCs line has distinct differentiation biases. Authors found that the expression SALL3 is highly correlated with the neuroectodermal differentiation potential but negatively correlated with mesodermal differentiation potential of hiPSCs. Authors used overexpression and depletion of SALL3 to demonstrate the function of SALL3 in neuroectoderm differentiation. Their findings are interesting, but preliminary. The function of Sall3 in neuroectoderm development has been reported before (Parrish, et al, MCB, 2004), and function of SALL3 in inhibition of DNA methylation through binding with DNMT3A was investigated in hepatocellular carcinoma (Shikauchi, et al, 2009). Although the focus of current manuscript is to determine the marker for biased differentiation potential of hiPSCs, statistical power is limited. Despite the weakness of the manuscript, manuscript still presents important concept of characterizing the hiPSCs based on expression of a few selected markers, and describes the function of SALL3 in neuroectoderm development in human embryonic context. Below are comments to improve the manuscript.

Response: We thank the reviewer for the positive remarks.

Major comments

1. Using the 10 hiPSC lines to identify the SALL3 as a marker for higher neuroectodermal differentiation potential is impressive. However, R-12A in current study expresses high level of SALL3 (Figure 2B) but show less neural differentiation (Figure C). Thus, the expression level of SALL3 is not absolute determinant of neuroectodermal potential. Authors should perform meta analysis including other hiPSC or hESC studies that performed gene expression and comparison of differentiation potential to demonstrate that SALL3 expression implicates the biased neuroectoderm differentiation potential in a statistically significant manner. Studies by Koyanagi-Aoi, et al, PNAS, 2013; Salomonis, et al, Stem cell Reports, 2016; Nishiazawa, et al, Cell Stem Cell, 2016 among others should be analyzed to show the positive or negative correlation of SALL3 expression from neuroectoderm or mesoderm differentiation.

Response: We thank the reviewer for the suggestion. We agree with the importance of

confirming the generality of our study using meta data. We carefully checked their studies and tried to demonstrate that *SALL3* expression implicates the differentiation propensity with a statistical significance. Unfortunately, only a part of their data was deposited in the public database and did not meet the criteria of our analysis. Instead, as our response to the comment 2 of reviewer 1#, to confirm our hypothesis that *SALL3* could be a marker for differentiation propensity, we additionally employed other two hiPSC lines, 606A1 and 648A1, as a test set and differentiated these hiPSCs into EBs. As expected from the *SALL3* expression levels in hiPSC lines, EBs derived from 606A1 hiPSCs exhibited higher expression of ectoderm marker genes, and lower expression of both mesoderm marker genes and endoderm marker genes, compared with 648A1 hiPSCs (Fig. S2).

2. hiPSCs are highly heterogeneous and their transcriptome changes during serial passage, because of epigenetic memory (Kim et al., Nature, 2010), aberrant epigenetic DNA methylation (Weissbein et al., PLoS Genetics, 2017) and X chromosome erosion (Mekhoubad et al., 2012; Nazor et al., 2012)). Therefore, parental cell, exact passage number and gender are key factor to produce differentiation biases in hiPSCs. Authors need to describe and examine whether the differentiation biases that they observed are dependent on these factors or not.

Response: We thank the reviewer for raising the point. We used 10 hiPSC lines for the statistical analysis of differentiation propensity, and these hiPSC lines have different backgrounds such as parental cells, passage number, gender, reprogramming methods. Even though the used hiPSC lines varied in many background factors, *SALL3* was identified as a key molecule to predict differentiation propensity of hiPSCs. Because seeking the background factors related to differentiation biases is out of the scope in this study to efficiently select suitable cell lines for products in many cell lines having various backgrounds, we have not considered the background factors of hiPSCs in our analysis. We would like to add a table showing cell line information including their backgrounds when used in the experiment (Table S2).

3. Lineage commitment of cells in EB is highly dependent on their environment. In addition, authors evaluated only differentiation potential by in vitro differentiation. To validate whether their differentiation potential is not dependent on medium or in vitro

conditions, they also should evaluate it by another assay, such as teratoma assay by overexpressing or knockdown SALL3 and injection.

Response: We thank the reviewer for raising the point. As the reviewer mentioned, we agree that various environmental factors will affect the differentiation propensity of hiPSCs. When identifying a predictive marker for hiPSC differentiation propensity, it is ideal to evaluate the differentiation propensity following the established differentiation protocol to manufacture the product. The teratoma assay has been reported to be only qualitative and not to be suitable for quantitatively assessing the extent of differentiation (Bouma et al. *Stem Cell Reports* (2017) 8, 1340-53). Indeed, we have previously experienced that hiPSCs subcutaneously injected into immunodeficient mice markedly vary in their differentiation capability among mice even though the same hiPSC line is transplanted. Therefore, we feel it is beyond the scope of this study and would like to reconsider the comment in the future work.

4. Several groups reported that gene body DNA methylation plays active role in transcription (Ball et al., 2009). However, hypermethylation on gene body in Sall3 KD was observed in neuronal-related genes, which were downregulated in Sall3 KD. Authors should explain this contradiction. Authors should perform the correlation of gene body or promoter methylation affected by SALL3 knockdown, and perform validation experiments, such as qPCR.

Response: We thank the reviewer for the suggestion. As described in response to comment 3 of reviewer 2#, we examined the comprehensive transcriptional profiles of wild type and SALL3 KD hiPSCs using microarray analysis (Dataset 5). We were not able to find any difference in the mRNA expression of genes directly related with ectoderm differentiation between the undifferentiated control and SALL3 KD cells for the purpose of explaining ectoderm differentiation propensity of hiPSCs. These suggest that gene body hypermethylation in neuronal-related genes do not contribute to their expression in the undifferentiated states. Although these results also suggest an importance of epigenetic changes in undifferentiated hiPSCs, of which gene expression would change accompanied with differentiation, epigenetic changes of gene body in neuronal-related genes by SALL3 seem not to influence the differentiation propensity of

hiPSCs in our study. In addition, we could not find any changes of DNA methylation in the promoter regions of neuronal-related genes such as PAX6 by SALL3 KD. The pathway analysis of epigenetically modified genes has suggested that SALL3 KD modulates the network of gene methylation in cell signaling, especially WNT-related signal, in hiPSCs (Table S1). Therefore, we prefer the model that SALL3 mainly influence cell signal pathways pivotal in embryonic differentiation to determine the hiPSC differentiation propensity.

5. Since their ChIP-seq does not show significant difference, SALL3 may regulate DNA methyltransferase activity of DNMT3B rather than DNA binding property. Therefore, I propose DNMT activity assay experiment in WT and SALL3-depleted cells to dissect SALL3-mediated inhibition of DNA methylation. Also, authors need to examine whether the binding sites of SALL3 is correlated with change in DNA methylation or DNMT3B binding sites.

Response: As described in response to the comments 4 of reviewer 1#, we established SALL3 knockout cells and measured DNMT activity of the nuclear fraction prepared from the cells. The knockout of *SALL3* gene markedly increased endogenous DNMT activity of hiPSCs, suggesting the inhibitory role of SALL3 in molecular function of DNMTs (Fig. 5C). Unfortunately, the quantitative measurement of the endogenous DNMT3B isoform activity after immunoprecipitation with anti-DNMT3B antibody was not successful. Therefore, to clarify that SALL3 selectively bind to DNMT3B but not other DNMTs contributing DNMT activity, we additionally demonstrate that SALL3 failed to interact with a maintenance enzyme DNMT1 as well as DNMT3A in hiPSCs (Fig. 5A). As described in response to the comment 5 of reviewer 1# and the comment 4 of reviewer 2#, we checked the DNA binding of SALL3 by ChIP-seq analysis. According to the ChIP-seq data, SALL3 was broadly enriched at CpG islands on genome. We could also find that SALL3 KD enhanced recruitment of DNMT3B to some of CpG island regions of gene body in WNT3A, WNT5A, BMP4 and DHH genes associated with the hypermethylation and SALL3 enrichment (Fig. 5G and Fig. S9).

Minor comments

1. Description of bioinformatics analysis is limited and it is hard to understand how

authors performed the analysis. More detail is needed. For example, R software has multiple functions for hierarchical clustering (in 352 line in page 23). Authors need to describe which library they used and what parameters they used. Version information of software is also needed.

Response: We thank the reviewer for pointing out a missing part of the methods. We have added information of R software in the Material and Methods section.

2. Heatmap requires “gradient color key” rather than simple color explanation. Authors also need to describe what heat colors represent (raw microarray intensity, z-score or any normalized values?) rather than “gene expression”.

Response: We thank the reviewer for carefully checking the figures and texts. We have corrected them accordingly (Fig. 1B).

Finally, we edited Fig. 2A and created Fig. 5A, C and G, Figs. S2, S7 and S9, Dataset S5 and Table S2, which were added to the revised manuscript.

Reviewers' comments:

Reviewer #1 (Remarks to the Author):

The revised manuscript has addressed prior criticisms and the text has improved. I am in favor of publication.

Reviewer #2 (Remarks to the Author):

The revised manuscript has addressed some of the issues pointed out in the first revision, but they could not address the molecular mechanism for the abnormality of neural differentiation in SALL3 KD hiPSCs. The authors claim that the experiment of Dnmt3B KD is difficult to rescue the phenotype of SALL3 KD hiPSCs. However, the phenotype of defect of neural differentiation in TET mutant ESCs is partially rescued by Dnmt3B KO. Therefore, if the abnormal methylation by SALL3 KD is caused for the defect of neural differentiation, DNMT3B KO or KD should rescue the phenotype of SALL3 KD. They could also try the site-specific DNA demethylation using dCas9/TET system in WNT3/5A locus to clarify the molecular mechanism for the abnormality of neural differentiation in SALL3 KD hiPSCs. I can't recommend the revised manuscript will be published in Nature Communications due to lack of the molecular mechanism.

Reviewer #3 (Remarks to the Author):

Authors have addressed most of comments, and the manuscript is ready to publish.

Response to reviewers:

Reviewer #2 (Remarks to the Author)

The revised manuscript has addressed some of the issues pointed out in the first revision, but they could not address the molecular mechanism for the abnormality of neural differentiation in *SALL3* KD hiPSCs. The authors claim that the experiment of *Dnmt3B* KD is difficult to rescue the phenotype of *SALL3* KD hiPSCs. However, the phenotype of defect of neural differentiation in TET mutant ESCs is partially rescued by *Dnmt3B* KO. Therefore, if the abnormal methylation by *SALL3* KD is caused for the defect of neural differentiation, *DNMT3B* KO or KD should rescue the phenotype of *SALL3* KD. They could also try the site-specific DNA demethylation using dCas9/TET system in *WNT3/5A* locus to clarify the molecular mechanism for the abnormality of neural differentiation in *SALL3* KD hiPSCs. I can't recommend the revised manuscript will be published in Nature Communications due to lack of the molecular mechanism.

Response: We would like to thank you for evaluating our manuscript (NCOMMS-17-27883B). We believe that your essential concern is whether the *DNMT3B* KO or KD should rescue the phenotype of *SALL3* KD. To answer this concern, we tried to generate *SALL3/DNMT3B* double-knockout lines. However, despite identifying the biallelic mutations in *DNMT3B* gene, we were unable to obtain any homozygous *DNMT3B* knockout hiPSC clones. Therefore, among the clones obtained, we employed *SALL3*^{-/-} *DNMT3B*^{-mut} clone, which possesses only a heterozygous *DNMT3B* gene with a single nucleotide substitution and an eighteen base addition mutation in the *SALL3* knockout background (Fig. S7A), for further experiments. The introduction of *DNMT3B*^{-mut} showed partial rescue of the attenuated neural differentiation observed in *SALL3*^{-/-} cells, when we examined the mRNA expression of *PAX6* and *SOX1* (Fig. 5D). Furthermore, *SALL3*^{-/-} *DNMT3B*^{-mut} cells significantly suppressed the enhanced cardiomyocyte differentiation of *SALL3*^{-/-} cells accompanied with the change of the expression of *GATA4*, *NKX2.5* and *TNNT2* (Fig. 5E). Our results suggest that *DNMT3B* is functionally contributes to the lineage differentiation phenotypes observed in *SALL3*-depleted hiPSCs.

We agree with the reviewer that the site-specific DNA demethylation using dCas9/TET system in *WNT3A/5A* locus is important experiments to clarify the

molecular mechanism for the abnormality of neural differentiation in SALL3 KD hiPSCs. We have showed that SALL3-depletion affects many important genes known to be involved in cell signaling of differentiation. We cannot exclude the possibility that other genes such as BMP4 and DHH also concomitantly participate in the regulation, which will need further careful protocol optimization and cell characterization. We feel it is beyond the scope of this study and would like to test it in future work.

We have revised the results (page 17, line 231 - page 18, line 253), discussion (page 24 line 343 – page 25, line 348), materials and methods (page 38, line 559 - page 39, line 577), and figure legend (page 54, line 782 – page 58, line 792). We assigned previous Figs. 5D-G to Figs. 6A-D because of adding new Figs. 5D and 5E to our manuscript. We also prepared Fig. S7.

REVIEWERS' COMMENTS:

Reviewer #2 (Remarks to the Author):

I am satisfied the authors have responded all the requested revisions from the previous review.